# Are We on the Right Way for Evaluating Large Vision-Language Models?

**Lin Chen**[1,3*] **Jinsong Li**[2,3*] **Xiaoyi Dong**[2,3] **Pan Zhang**[3] **Yuhang Zang**[3]
**Zehui Chen**[1] **Haodong Duan**[3] **Jiaqi Wang**[3†] **Yu Qiao**[3] **Dahua Lin**[2,3,4] **Feng Zhao**[1†]
[1] University of Science and Technology of China
[2] The Chinese University of Hong Kong
[3] Shanghai AI Laboratory    [4] CPII under InnoHK
`https://mmstar-benchmark.github.io/`

## Abstract

Large vision-language models (LVLMs) have recently achieved rapid progress, sparking numerous studies to evaluate their multi-modal capabilities. However, we dig into current evaluation works and identify two primary issues: 1) **Visual content is unnecessary for many samples.** The answers can be directly inferred from the questions and options, or the world knowledge embedded in LLMs. This phenomenon is prevalent across current benchmarks. For instance, GeminiPro achieves 42.7% on the MMMU benchmark *without* any visual input, and outperforms the random choice baseline across six benchmarks near 24% on average. 2) **Unintentional data leakage exists in LLM and LVLM training.** LLM and LVLM could still answer some visual-necessary questions without visual content, indicating the memorizing of these samples within large-scale training data. For example, Sphinx-X-MoE gets 43.6% on MMMU *without* accessing images, surpassing its LLM backbone with 17.9%. Both problems lead to misjudgments of actual multi-modal gains and potentially misguide the study of LVLM. To this end, we present **MMStar**, an elite vision-indispensable multi-modal benchmark comprising 1,500 samples meticulously selected by humans. MMStar benchmarks 6 core capabilities and 18 detailed axes, aiming to evaluate LVLMs' multi-modal capacities with carefully balanced and purified samples. These samples are first roughly selected from current benchmarks with an automated pipeline, human review is then involved to ensure each curated sample exhibits visual dependency, minimal data leakage, and requires advanced multi-modal capabilities. Moreover, two metrics are developed to measure data leakage and actual performance gain in multi-modal training. We evaluate 16 leading LVLMs on MMStar to assess their multi-modal capabilities, and on 7 benchmarks with the proposed metrics to investigate their data leakage and actual multi-modal gain.

## 1 Introduction

Encouraged by the rapid development of large language models (LLMs) [60, 4, 9, 10, 14, 1, 53], integrating visual modality into LLMs to enhance models' interactivity capabilities has witnessed ever-changing advances in recent days [72, 33, 31, 12, 68, 2, 61, 39, 5, 13]. These large vision-language models (LVLMs) showcase powerful visual perception and understanding capabilities, enabling them to accept image inputs from users and engage in dialogues, thereby offering a more enriched interactive experience. These achievements have further inspired the research community

---

*Equal contribution. This work is done during internship in Shanghai AI Laboratory.
†Corresponding author.

38th Conference on Neural Information Processing Systems (NeurIPS 2024).

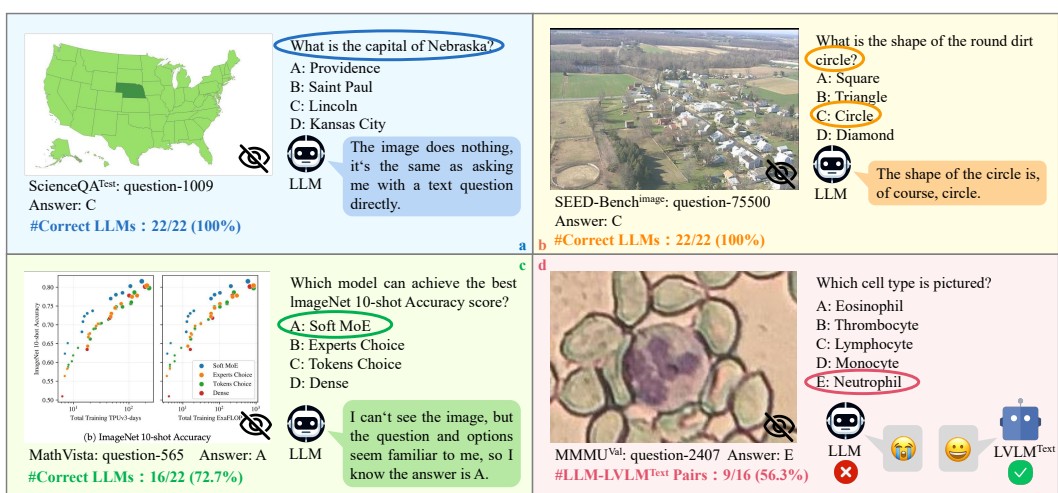

Figure 1: We highlight cases in existing multi-modal benchmarks where evaluation samples either **lack visual dependency** or **have unintentionally leaked into the training data of LLMs and LVLMs**. (a) Some samples can be answered by LLMs using only text-based world knowledge; (b) For some instances, the question itself contains the answer, making images superfluous; (c) Some samples are leaked into LLMs' training corpora can be "recalled" with the textual questions and answers directly; (d) Some samples indiscernible to LLMs but solved by LVLMs without accessing images suggest leakage into LVLMs' multi-modal training data.

to develop a variety of multi-modal benchmarks [27, 16, 34, 47, 63, 64, 37, 26, 38], constructed to explore the powerful capabilities emerging from LVLMs and provide a comprehensive and objective platform for quantitatively comparing the continually evolving models. Despite the race among existing evaluation works to construct as many axes as possible to assess the capabilities of LVLMs, we have identified two primary issues upon delving into existing evaluation samples and processes.

First, **visual content is unnecessary for many samples**. A qualified multi-modal evaluation sample should compel LVLMs to understand and reason with the visual content for correct answers. Otherwise, the evaluation sample would degrade into assessing the textual capabilities of LLM bases. Unfortunately, we have identified numerous samples across multiple popular benchmarks [34, 27, 64, 38, 26] where answers can be correctly deduced without relying on visual content. As shown in Figure 1 (a) and (b), some samples have answers directly included within the questions (e.g., What is the shape of the round dirt circle?), while others can be effortlessly answered by leveraging the rich world knowledge embedded within the LLM bases (e.g., What is the capital of Nebraska?). As shown in Table 1, with a comprehensive quantitative analysis of 22 LLMs on 6 benchmarks, we observe this phenomenon is prevalent and serious. For example, more than 50% questions of ScienceQA and near 30% questions of MMMU can be solved by most LLMs directly. For the powerful LLM GeminiPro, it achieves 42.7% on the MMMU benchmark without any visual input, and outperforms the random choice baseline across six benchmarks by near 24% on average.

Taking aside the inappropriate samples in evaluation, we also observed strange results that LLM and LVLM could still answer some visual-necessary questions without visual content (Figure 1 (c) and (d)). A plausible explanation for this could be the inadvertent memorization of these samples during the large-scale training process, suggesting the presence of **unintentional data leakage in the training of LLM and LVLM**. Through a detailed study of various LVLMs on 6 benchmarks, as shown in Table 2, we find the unexpected leaking problem during the LVLM training is particularly serious. For example, we find Yi-VL-34B gets 15.0% higher performance than its LLM backbone on ScienceQA, Sphinx-X-MoE gets 43.6% on MMMU *without* accessing images, surpassing its LLM backbone with 17.9%, even surpassing many leading LVLMs with accessing images.

The existence of inappropriate questions and data leaking would lead to misjudgments of actual multi-modal performance gains and potentially misguide the study of LVLM. In pursuit of a more accurate and comprehensive evaluation, we introduce the MMStar Benchmark. MMStar is a premier, vision-critical multi-modal benchmark that includes 1,500 challenging samples, each rigorously validated by humans. It is structured to test 6 fundamental capabilities and 18 specific di-

mensions, aiming to evaluate the multi-modal capacities of LVLMs with a carefully balanced and purified selection of samples.

The MMStar is a new benchmark that "Stands on the shoulders of giants". Samples are first roughly selected from current benchmarks with an automated pipeline. In detail, we use eight powerful LLMs as candidates inspectors for visual dependency and LLM leakage, including two closed-source APIs (GPT4-Turbo [42], and GeminiPro [51]) and six leading open-source models (e.g., LLaMA-70B [53], Qwen-1.5-72B [1], and Mixtral-8x7B [23]). Samples that could be answered by more than 2 of the 8 LLMs are excluded as they may exist leaking or visual-unnecessary problems. Then we use 16 leading LVLMs (e.g., GPT4V [43], GeminiPro (Vision) [51], LLaVA series [31, 33]) to gauge the difficulty of the samples and split them to four levels. Ultimately, based on the difficulty of the rough-filtered samples, **strict manual review and selection** are applied to curate 1,500 high-quality multimodal evaluation samples. As shown in Figure 3, these samples span 6 core multimodal capability dimensions and 18 detailed axes, aiming to probe LVLMs' advanced multimodal capabilities with a purified and high-quality set of samples. Moreover, we design the multi-modal gain (MG) and multi-modal leakage (ML) metrics to probe LVLMs' actual performance gain and data leakage degrees derived from multi-modal training in a benchmark-specific manner.

We evaluate the accuracy, MG, and ML of 16 leading LVLMs on our MMStar benchmark, the high-resolution version of GPT-4V ranks first with 57.1% accuracy, showcasing its superb multi-modal capability. GPT-4V also gets the best MG and a small ML, indicating its effective multi-modal training strategy and has less data leaking.

In a nutshell, our contributions are threefold:

- We delve into existing evaluation benchmarks and processes and identify two key issues: (1) Visual content is unnecessary for many samples. (2) Unintentional data leakage exists in LLM and LVLM training. Both lead to misjudgment of LVLM and may misguide the following study.
- We curate MMStar, an elite vision-indispensable multi-modal benchmark comprising 1,500 challenge samples meticulously selected by humans. MMStar covers samples from diverse tasks and difficulties, aiming to evaluate the actual multi-modal capacities of LVLMs.
- Based on MMStar, we evaluate LVLMs with Accuracy and two newly proposed metrics: multi-modal gain and multi-modal leakage. The high-resolution version of GPT-4V outperforms the 16 leading LLMs and ranks first.

## 2   Related Work

**Large Vision-Language Models.** As large language models (LLMs) [9, 53, 53, 60, 52, 42, 65, 44, 10] rapidly advance, a growing fraction of the research community is focusing on integrating visual content into LLMs to build a powerful intelligent assistant with more interactive ways. Central to these large vision-language models (LVLMs) are the seminal works in modality alignment within the vision-language learning area [46, 21]. The foundation work CLIP [46] exemplifies the alignment of vision and language modalities through contrastive learning on extensive image-text pairs. Built upon the CLIP image encoder which is somewhat aligned with the language modality, current LVLMs typically utilize vast image-text pairs to connect the vision encoder and LLM, enabling LLM to receive and understand visual content [72, 33, 31, 12, 45, 2, 48, 6, 39, 19, 5, 20, 71, 73, 24, 59, 70]. For example, MiniGPT4 [72] and LLaVA [33] directly connect the vision encoder and LLM with QFormer [28] and MLP [50], showing proficiency in multi-modal dialogues. Subsequent works have further enhanced LVLMs by improving the multi-modal instruction data [31, 61, 5, 54, 67, 25] and designing novel modules [2, 30, 55, 36, 17, 13] for more sufficient modality alignment.

**Evaluations of LVLMs.** To probe the true capabilities of the emerging LVLMs, the research community has developed many multi-modal benchmarks encompassing a wide range of evaluation axes [34, 16, 47, 64, 49, 66, 27, 33, 29, 35, 63, 56, 58, 57]. Early single-task benchmarks, such as VQA [18], MS-COCO [49], and OK-VQA [47], fail to holistically assess LVLMs' general multi-modal perception and reasoning capabilities. To address this issue, comprehensive multi-modal benchmarks have been constructed [33, 27, 64, 16, 34, 8, 56]. For example, SEED [27] and MMBench [34] cover 12 and 20 evaluation dimensions respectively, while MMMU [64] spans 30 college-level subjects, providing some competitive arenas for a comprehensive comparison of cutting-edge LVLMs. However, existing evaluations of LVLMs overlook some critical issues. On the one hand, they do not guarantee that all evaluation samples cannot be correctly answered without the visual

Table 1: **Evaluation of various LLMs on six popular multi-modal benchmarks under 2-shot.** We employ a 2-shot inference strategy for evaluating all LLMs to reduce instances of refusal to answer and align the answer formats. We report the results of 2 closed-source LLMs and 20 open-source LLMs with varying sizes and architectures. The evaluated benchmarks include MMMU (MMMU-Val [64]), MMB (MMBench-EN-Dev [34]), ScienceQA (ScienceQA-Test [38]), AI2D (AI2D-Test [26]), SEED (SEED-Image [27]), and MathVista (MathVista-Mini [37]). The **best** results are highlighted in **bold and underlined.**

| Model | Strategy | MMMU | MMB | ScienceQA | AI2D | SEED | MathVista | Avg. |
|---|---|---|---|---|---|---|---|---|
| | | | | *Baseline* | | | | |
| Random Choice | - | 22.1 | 0.0 | 24.2 | 23.8 | 24.3 | 17.9 | 18.7 |
| | | | | *Closed-source LLMs* | | | | |
| GPT4-Turbo[42] | 2-shot | 42.0 | 15.5 | 67.5 | **61.3** | 26.8 | **25.6** | 39.8 |
| GeminiPro[51] | 2-shot | **42.7** | **18.7** | **69.3** | 60.1 | **38.1** | 25.5 | **42.4** |
| | | | | *Open-source LLMs* | | | | |
| Qwen1.5-1.8B[1] | 2-shot | 33.0 | 8.6 | 55.6 | 41.3 | 32.1 | 22.7 | 32.2 |
| Phi2-2.7B[40] | 2-shot | 19.9 | 4.3 | 50.8 | 41.7 | 6.9 | 18.4 | 23.7 |
| Yi-6B[62] | 2-shot | 32.9 | 16.0 | 64.6 | 51.5 | 36.7 | 24.5 | 37.7 |
| LLaMA2-7B[53] | 2-shot | 25.9 | 7.7 | 57.9 | 42.8 | 32.8 | 22.8 | 31.7 |
| Qwen-7B[1] | 2-shot | 30.6 | 15.0 | 63.0 | 50.0 | 32.6 | 21.0 | 35.4 |
| Deepseek-7B[3] | 2-shot | 28.7 | 11.6 | 61.9 | 46.0 | 34.1 | 21.7 | 34.0 |
| InternLM2-7B[52] | 2-shot | 33.6 | 11.4 | 63.6 | 52.1 | 34.4 | 20.4 | 35.9 |
| Qwen1.5-7B[1] | 2-shot | 33.3 | 13.1 | 65.1 | 52.1 | 32.1 | 22.8 | 36.4 |
| Vicuna-v1.5-7B[9] | 2-shot | 31.3 | 9.5 | 58.9 | 45.5 | 32.0 | 20.7 | 33.0 |
| Baichuan2-7B[60] | 2-shot | 28.2 | 13.7 | 58.1 | 44.1 | 32.3 | 21.7 | 33.0 |
| Mistral-7B[22] | 2-shot | 29.8 | 17.2 | 66.1 | 50.0 | 34.4 | 13.4 | 35.2 |
| LLaMA2-13B[53] | 2-shot | 32.9 | 10.1 | 58.9 | 43.8 | 32.1 | 24.8 | 33.8 |
| Vicuna-v1.5-13B[9] | 2-shot | 31.3 | 12.8 | 63.0 | 46.8 | 33.6 | 20.8 | 34.7 |
| Baichuan2-13B[60] | 2-shot | 32.2 | 13.1 | 61.0 | 47.1 | 35.2 | 23.4 | 35.3 |
| InternLM2-20B[52] | 2-shot | 35.6 | 17.4 | 66.4 | 55.9 | 30.4 | 20.8 | 37.8 |
| Yi-34B[62] | 2-shot | 35.8 | 15.8 | 67.9 | 59.6 | 37.2 | **26.9** | 40.5 |
| Mixtral-8x7B[23] | 2-shot | 35.1 | 17.3 | 66.3 | 55.1 | 35.8 | 22.7 | 38.7 |
| Deepseek-67B[3] | 2-shot | 38.3 | 17.2 | 68.3 | 59.7 | 37.3 | 23.4 | 40.7 |
| LLaMA2-70B[53] | 2-shot | 30.4 | 17.2 | 63.4 | 49.3 | 34.9 | 24.2 | 36.6 |
| Qwen1.5-72B[1] | 2-shot | **42.4** | **21.1** | **70.1** | **60.9** | **40.7** | 26.3 | **43.6** |

content. On the other hand, current evaluations consistently adhere to the process of inferring on given benchmarks and calculating scores for LVLMs, overlooking the possibility of data leakage during multi-modal training. This oversight can lead to unfair comparisons and misjudgments of the real gains in multi-modal capabilities brought by multi-modal training.

## 3   Two Overlooked Issues for Evaluating LVLMs

In this section, we delve into two commonly overlooked issues in current LVLM evaluation works. Moreover, we present detailed experimental results to further substantiate our observations.

**First issue: visual content is unnecessary for many evaluation samples.** The key distinction between evaluating LLMs and LVLMs lies in the necessity for LVLM evaluations to strictly ensure that the correct answers can only be derived based on a thorough understanding of visual content. Without this, evaluating LVLMs' multi-modal capabilities degrades to merely assessing their LLM backbones' uni-modal abilities. However, upon examining samples from some popular LVLM benchmarks, we find many samples lack vital visual dependency and can yield correct answers even without the image inputs! Through analysis of these failure samples, we categorize them into two groups: (1) Answers can be directly obtained from the world knowledge embedded in LLMs, owing to the LLMs' extensive pretraining on the large corpus of data. For example, as illustrated in Figure 1(a), the question "What is the capital of Nebraska?" already provides the key information "Nebraska", eliminating the need for extracting relevant location information from visual content. A more appropriate question is "What is the capital of the highlighted area in the image?" to emphasize

the importance of visual understanding. (2) Answers are directly included in the textual questions. As shown in Figure 1(b), LLMs can derive the correct answer "circle" through simple reasoning based on the question "What is the shape of the round dirt circle?".

To quantitatively substantiate our findings, we further experiment to gauge the proportion of these two types of samples in existing benchmarks. Specifically, we evaluate several benchmarks with two closed-source LLMs (GPT4-Turbo [42], and GeminiPro [51]) and six open-source heavy LLMs (InternLM2-20B [52], Yi-34B [62], Mixtral-8x7B [23], Deepseek-67B [3], LLaMA2-70B [53], and Qwen1.5-72B [1]), recording the hit count for each question. Here, the 'hit' refers to the ability of an LLM to cor-

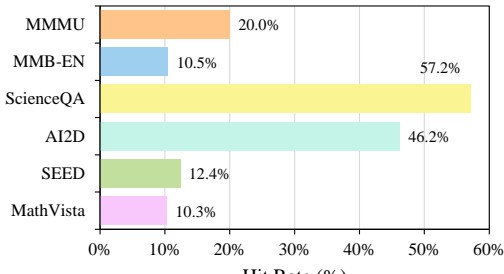

Figure 2: LLM hit rate across various benchmarks.

rectly answer the question without relying on visual input. We then calculate the percentage of samples with a hit count of six or more (80%) against the total number of samples to determine the abnormal hit rate for each benchmark. As depicted in Figure 2 , every benchmark shows a certain degree of samples that visual contents are unnecessary, with ScienceQA [38] and AI2D [26] exhibiting amazing abnormal hit rates of 57.2% and 46.2%, respectively. Based on our observations, most multi-modal benchmarks have yet to fully assess the multi-modal capabilities of LVLMs.

**Second issue: unintentional data leaking exists in LLM and LVLM training.** Although the community has the trend towards developing new multi-modal benchmarks to assess LVLMs' capabilities from various dimensions, there is scant consideration for fairness and reliability during evaluation. Training LLMs and LVLMs requires vast and diverse data, inevitably leading to the leakage of evaluation samples. Such incidents are usually unintended, as it's impractical to predict which data will be used in future evaluation benchmarks during the preparation for training corpus.

Figure 1 (c) showcases an evaluation sample leaked by LLMs. Though the question requires an understanding of image content, 16 out of 22 tested LLMs astonishingly provide the correct response by "recalling" their training data. To quantitatively support our observations, we evaluate 22 leading LLMs across 6 popular benchmarks and report the 2-shot results in Table 1. Specifically, we find the 2-shot evaluation strategy is more stable than the 0-shot (see results in Section A.8) to reduce refusal for answering and align answer formats. Under the impact of vision-independent samples and data leakage from LLMs, GeminiPro [51] and Qwen1.5-72B [1] achieve a remarkable average accuracy of 42.4% and 43.6% under the 2-shot setting, outperforming random choice by 21.4% and 22.6%, respectively. Furthermore, Qwen1.5-72B achieves a score of 42.4% on MMMU [64], even surpassing the performance of the majority of LVLMs with accessing images. This result serves as a reminder: if we only consider the final accuracy on benchmarks when evaluating LVLMs, potential data leakage from LLMs could lead to unfair comparisons.

In Figure 1 (d) and Section A.5, we showcase some examples where original LLMs fail, but LVLMs without accessing images succeed. Despite these questions requiring image content for accurate answers, the LVLMs without accessing images are capable of correctly answering these questions which stump original LLMs. To further support our hypotheses of data leakage during LVLMs' multi-modal training, we conduct an intriguing experiment. We remove the images for LVLMs and only utilize questions and options for evaluation, with results reported in Table 2. We compare the gains of LVLMs set to receive only text inputs (LVLM-text) against their corresponding LLM bases (LLM) to quantitatively assess the degree of data leakage in LVLMs' multi-modal training. As shown in Table 2, most LVLMs exhibit varying degrees of data leakage during multi-modal training. For example, the LLMs of Sphinx-X-8x7B [17] and Monkey-Chat [30], show a respective average performance gain of 14.1% and 14.2% compared to their original LLMs.

Drawing from our observations, we posit that the issue of data leakage in multi-modal datasets is a significant concern that warrants attention. Addressing this issue is essential to ensuring that model performance is measured by their genuine ability to integrate and interpret multimodal data, rather than by their tendency to memorize specific samples within the dataset. Establishing a robust and reliable benchmark to minimize data leakage would thus serve as a foundational step in advancing research within the field of multimodal language models, paving the way for more meaningful and accurate evaluations of their performance and potential.

Table 2: **Evaluation of various LVLMs on six popular multi-modal benchmarks.** For the "strategy" column, "LLM" refers to evaluating using the corresponding LLM base of the LVLM, while "LVLM-text" denotes evaluating LVLMs without accessing images. We employ the **0-shot** inference strategy for LLMs to align the evaluation protocols of LVLMs. We only report the results of 2 closed-source LVLMs and 8 open-source LVLMs due to space limits. For the entire LVLMs' results, please refer to the appendix. The **highest** results of the LVLM-text setting across the models are highlighted in **bold and underlined.**

| Model | Param. | Strategy | MMMU | MMB | ScienceQA | AI2D | SEED | MathVista | Avg. |
|---|---|---|---|---|---|---|---|---|---|
| *Baseline* | | | | | | | | | |
| Random Choice | - | - | 22.1 | 0.0 | 24.2 | 23.8 | 24.3 | 17.9 | 18.7 |
| *Closed-source LVLMs and corresponding LLM bases* | | | | | | | | | |
| GPT4V[43] | - | LLM | 41.2 | 12.2 | 64.3 | 59.7 | 10.1 | 24.2 | 35.3 |
| (GPT4-Turbo[42]) | - | LVLM-text | **45.1** | **17.6** | **68.2** | **62.5** | **28.4** | **25.4** | **41.2** |
| | - | LVLM | 53.6 | 69.6 | 81.4 | 75.3 | 71.6 | 44.7 | 66.0 |
| GeminiPro (Vision)[51] | - | LLM | 42.9 | 18.4 | 68.9 | 59.2 | 35.5 | 23.3 | 41.4 |
| (GeminiPro[51]) | - | LVLM-text | 39.4 | 16.7 | 66.3 | 54.5 | 27.9 | 24.5 | 38.2 |
| | - | LVLM | 44.4 | 68.1 | 80.6 | 68.0 | 64.3 | 36.0 | 60.2 |
| *Open-source LVLMs and corresponding LLM bases* | | | | | | | | | |
| TinyLLaVA[69] | 3B | LLM | 20.0 | 7.2 | 47.1 | 38.7 | 26.4 | 22.0 | 26.9 |
| (Phi2-2.7B[40]) | | LVLM-text | 30.0 | 21.0 | 62.3 | 51.9 | 37.2 | 23.5 | 37.7 |
| | | LVLM | 36.0 | 66.9 | 69.1 | 62.4 | 70.1 | 28.9 | 55.6 |
| LLaVA-1.5[31] | 7B | LLM | 29.9 | 10.3 | 58.9 | 42.5 | 32.6 | 22.0 | 32.7 |
| (Vicuna-v1.5-7B[9]) | | LVLM-text | 29.9 | 19.5 | 64.1 | 48.7 | 37.5 | 20.3 | 36.7 |
| | | LVLM | 34.4 | 65.0 | 68.7 | 55.6 | 65.6 | 23.6 | 52.2 |
| InternLM2-XC2[13] | 7B | LLM | 32.8 | 8.9 | 64.0 | 48.3 | 31.9 | 18.9 | 34.1 |
| (InternLM2-7B[52]) | | LVLM-text | 34.2 | **26.2** | **71.9** | 63.3 | 38.1 | **29.4** | 43.9 |
| | | LVLM | 41.7 | 79.6 | 96.7 | 81.4 | 74.9 | 57.4 | 72.0 |
| Monkey-Chat[30] | 10B | LLM | 19.8 | 8.4 | 52.7 | 42.6 | 7.6 | 20.5 | 25.3 |
| (Qwen-7B[1]) | | LVLM-text | 32.4 | 15.6 | 71.1 | 56.8 | 36.1 | 25.0 | 39.5 |
| | | LVLM | 37.1 | 71.0 | 82.4 | 68.5 | 69.1 | 34.0 | 60.4 |
| CogVLM-Chat[55] | 17B | LLM | 29.9 | 10.3 | 58.9 | 42.5 | 32.6 | 22.0 | 32.7 |
| (Vicuna-v1.5-7B[9]) | | LVLM-text | 30.1 | 15.5 | 54.6 | 52.5 | 36.7 | 25.0 | 35.7 |
| | | LVLM | 34.2 | 63.4 | 66.3 | 63.3 | 68.7 | 34.7 | 55.1 |
| Yi-VL[62] | 34B | LLM | 37.1 | 10.5 | 53.6 | 57.3 | 37.3 | 21.7 | 36.3 |
| (Yi-34B[62]) | | LVLM-text | 37.3 | 23.2 | 68.6 | 59.9 | **41.0** | 22.7 | 42.1 |
| | | LVLM | 43.2 | 71.5 | 75.3 | 65.9 | 68.1 | 25.6 | 58.3 |
| InternVL-Chat-v1.2[7] | 40B | LLM | 37.6 | 20.1 | 69.4 | 60.2 | 35.0 | 17.9 | 40.0 |
| (NH2-Yi-34B[41]) | | LVLM-text | 41.7 | 23.9 | 70.3 | **65.0** | 40.5 | 24.0 | **44.2** |
| | | LVLM | 49.1 | 82.4 | 82.5 | 78.5 | 75.4 | 47.7 | 69.3 |
| Sphinx-X-MoE[17] | 57B | LLM | 25.7 | 8.6 | 57.2 | 48.7 | 13.5 | 23.4 | 29.5 |
| (Mixtral-8x7B[23]) | | LVLM-text | **43.6** | 20.5 | 68.4 | 61.1 | 39.9 | 28.4 | 43.7 |
| | | LVLM | 44.8 | 69.2 | 72.2 | 65.0 | 71.1 | 38.1 | 60.1 |

# 4 MMStar

## 4.1 Data Curation Process

**Criteria for data curation.** The evaluation samples for constructing the MMStar benchmark should meet three fundamental criteria: 1) **Visual dependency.** The collected samples can be correctly answered only based on understanding the visual content; 2) **Minimal data leakage.** The collected samples should minimize the risk of unintentional inclusion in LLMs' training corpus, or be effectively transformed from uni-modal to multi-modal formats to prevent LLMs from "recalling" the correct answers; 3) **Requiring advanced multi-modal capabilities for resolution.** In addition to ensuring fairness and reliability by adhering to the above criteria, we also aim for samples to cover various difficulty levels. We expect to comprehensively capture LVLMs' multi-modal capabilities with succinct high-quality samples.

**Data filter.** We first choose two benchmarks [34, 27] focused on natural images and four centered on scientific and technical knowledge [64, 38, 26, 37] for our sample collection. We then develop an automated pipeline to preliminarily filter out samples that do not meet the first two criteria. Specifically, we employ two closed-source LLMs [51, 42] and six open-source LLMs [1, 52, 62, 3, 23, 53] sizing 20B or larger to serve as inspectors. These open-source LLMs are applied with a

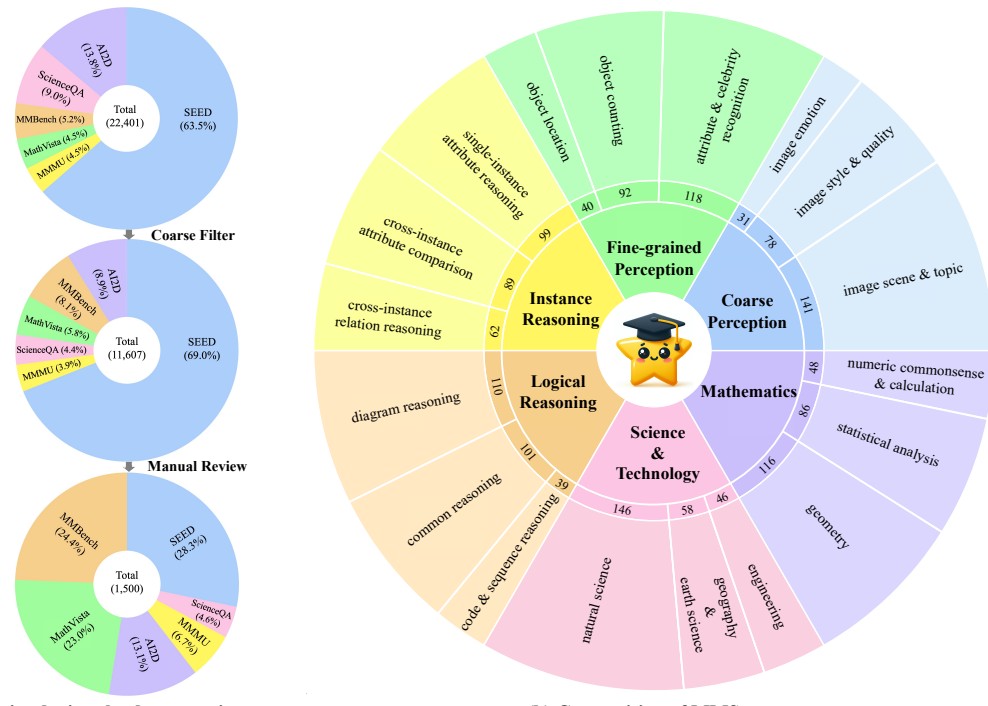

**(a) Statics during the data curation process**  **(b) Composition of MMStar**

Figure 3: **Details of MMStar benchmark.** (a) Statics of the data sources during the data curation process. After applying the coarse filter process and manual review, we narrow down from a total of 22,401 samples to 11,607 candidate samples and finally select 1,500 high-quality samples to construct our MMStar benchmark. (b) We display 6 core capabilities in the inner ring, with 18 detailed axes presented in the outer ring. The middle ring showcases the number of samples for each detailed dimension.

2-shot in-context inference strategy to minimize response refusals and ensure consistency in answer formatting. Following this, we evaluate the sample pool with these LLM inspectors, documenting the hit frequency for each evaluation sample. Finally, we only retain those samples with hit counts of two or fewer hits, indicating that around 75% of LLM inspectors fail to provide the correct answer. As illustrated in Figure 3 (a), following this initial coarse filtering, our sample pool was reduced from 22,401 to 11,607.

**Manual review.** After the coarse filtering with LLM inspectors, we further employ three experts to conduct the manual review process to ensure: 1) each sample's answer should be based on the understanding of visual content; 2) selected samples should cover a comprehensive range of capability assessment dimensions; 3) most samples should require LVLMs to possess advanced multi-modal abilities for resolution. To expedite the manual selection of samples with varying difficulty levels for LVLMs, we tally the hit counts of all 16 LVLMs on the coarsely filtered samples and split them into four difficulty categories: easy (12-16, 148 examples), moderate (8-11, 189 examples), hard (4-7, 631 examples), and tough (0-3, 532 examples). Finally, after considering both the diversity of capability dimensions and difficulty levels, we manually curated **1,500** high-quality samples from the coarsely filtered set. Figure 3 (a) showcases the detailed composition of data sources for our final selection of samples. In Section A.3, we provide details on how the manual review step aggressively reduces the MMStar benchmark from 11,607 samples to 1,500 samples.

## 4.2 Core Capabilities

We select and consolidate the dimensions used for assessing LVLMs' multi-modal capabilities in existing benchmarks and identify six core capabilities along with eighteen detailed axes. In Figure 3 (b), we provide statistics for each core capability and their detailed axes on the MMStar benchmark. **More detailed definitions of each capability are provided in Section A.2.**

Table 3: **Evaluation of various LVLMs on MMStar.** We report the results of 2 closed-source LVLMs and 14 open-source LVLMs with varying sizes and architectures. We report the detailed results of the CP (coarse perception), FP (fine-grained perception), IR(instance reasoning), LR (logical reasoning), ST (science & technology), and MA (mathematics) core capabilities. The **best** results are highlighted in **bold and underlined.** The *worst* results of multi-modal gain (MG) and multi-modal leakage (ML) metrics are in *italic red*.

| Model | LLM | Param. | CP | FP | IR | LR | ST | MA | Avg. | MG↑ | ML↓ |
|---|---|---|---|---|---|---|---|---|---|---|---|
| *Baselines* | | | | | | | | | | | |
| Random Choice | - | - | 23.7 | 24.5 | 25.3 | 24.3 | 24.8 | 25.1 | 24.6 | - | - |
| *Closed-source LVLMs* | | | | | | | | | | | |
| GeminiPro-Vision[51] | GeminiPro[51] | - | 51.6 | 28.8 | 50.8 | 46.0 | 28.4 | **50.0** | 42.6 | 27.4 | **0.0** |
| GPT4V (low)[43] | GPT4-Turbo[42] | - | 62.0 | 32.8 | 55.2 | 48.0 | 33.6 | 44.8 | 46.1 | 32.6 | 1.3 |
| GPT4V (high)[43] | GPT4-Turbo[42] | - | **76.6** | **51.4** | **66.6** | **55.8** | **42.6** | 49.8 | **57.1** | **43.6** | 1.3 |
| *Open-source LVLMs* | | | | | | | | | | | |
| TinyLLaVA[69] | Phi2-2.7B[40] | 3B | 60.4 | 31.6 | 50.8 | 30.4 | 18.0 | 24.8 | 36.0 | 16.4 | 7.6 |
| Yi-VL[62] | Yi-6B[62] | 6B | 58.0 | 33.6 | 46.4 | 34.8 | 20.4 | 34.0 | 37.9 | 15.6 | **0.0** |
| LLaVA-1.5[31] | Vicuna-v1.5-7B[9] | 7B | 58.8 | 24.0 | 38.8 | 24.0 | 13.6 | 22.8 | 30.3 | *10.7* | **0.0** |
| ShareGPT4V[5] | Vicuna-v1.5-7B[9] | 7B | 58.8 | 28.0 | 45.6 | 24.4 | 17.2 | 24.0 | 33.0 | 11.9 | **0.0** |
| InternLM-XC2[13] | InternLM2-7B[52] | 7B | **70.8** | 48.8 | **65.2** | **56.4** | **42.0** | 49.2 | **55.4** | 28.1 | 7.5 |
| Deepseek-VL[36] | Deepseek-7B[3] | 8B | 64.0 | 30.8 | 49.2 | 36.4 | 21.6 | 20.4 | 37.1 | 15.7 | **0.0** |
| Qwen-VL-Chat[2] | Qwen-7B[1] | 10B | 59.6 | 32.0 | 50.8 | 29.2 | 22.0 | 31.6 | 37.5 | 23.9 | **0.0** |
| Monkey-Chat[30] | Qwen-7B[1] | 10B | 57.6 | 36.4 | 51.6 | 33.2 | 26.4 | 24.4 | 38.3 | 13.5 | *17.6* |
| LLaVA-1.5[31] | Vicuna-v1.5-13B[9] | 13B | 58.8 | 28.0 | 41.6 | 24.4 | 18.4 | 25.6 | 32.8 | 13.9 | **0.0** |
| CogVLM-Chat[55] | Vicuna-v1.5-7B[9] | 17B | 66.8 | 36.8 | 49.2 | 31.2 | 23.6 | 11.6 | 36.5 | 14.9 | **0.0** |
| Yi-VL[62] | Yi-34B[62] | 34B | 53.2 | 31.2 | 52.0 | 32.4 | 12.4 | 35.2 | 36.1 | 18.8 | **0.0** |
| LLaVA-Next[32] | NH2-Yi-34B[41] | 34B | 66.4 | **52.0** | 62.4 | 46.0 | 32.4 | **53.6** | 52.1 | 29.4 | 2.4 |
| InternVL-Chat-V1.2[7] | NH2-Yi-34B[41] | 40B | 67.6 | 43.2 | 61.2 | 47.2 | 24.0 | 19.2 | 43.7 | **32.6** | **0.0** |
| Sphinx-X-MOE[17] | Mixtral-8x7B[23] | 57B | 58.4 | 40.8 | 47.6 | 35.2 | 19.2 | 32.0 | 38.9 | 14.8 | 1.0 |

## 4.3 Multi-modal Gain/Leakage

Given our observation of the potential for inadvertent leakage of some evaluation samples during the multi-modal training process, the vanilla evaluation approach struggles to reveal LVLMs' actual performance gains derived from multi-modal training and fails to enable fair comparison with other competitors. Therefore, we propose two novel metrics to separately assess the degree of data leakage and actual performance gain from the multi-modal training process.

To calculate the multi-modal gain (MG) metric for a given LVLM on a particular benchmark, we need to compute the scores of the same LVLM with and without visual inputs, separately denoted as $S_w v$ and $S_{ov}$. Then the MG metric can be derived from the following formulation:

$$MG = S_{wv} - S_{ov}. \tag{1}$$

To calculate the multi-modal leakage (ML) metric, we need to compute the extra score of the given LVLM's LLM base (without any multi-modal training), denoted as $S_t$. Then the ML metric is formulated as follows:

$$ML = \max(0, S_{ov} - S_t). \tag{2}$$

## 5 Experiments

In this section, we conduct a systematic analysis of the proposed MMStar benchmark along with the MG/ML metrics. These analyses encompass various LLMs and LVLMs, and also involve numerous existing benchmarks when examining MG/ML metrics. We choose VLMEvalKit [15] as our codebase. Please see details about experimental setups in Section A.1.

## 5.1 Results Analysis of MMStar

In this section, we present a comprehensive comparison of various LLMs and LVLMs performed on our MMStar benchmark and summarize our key observations in the following parts.

**Observation from LLMs.** We comprehensively evaluate 2 closed-source LLMs and 20 open-source LLMs of varying sizes and architectures on the MMStar benchmark and report the results in Figure 4 and Table 5. Encouragingly, the performance of these LLMs is almost indistinguishable from random choice, effectively validating that the evaluation samples of our MMStar exhibit significant visual dependency and minimal data leakage from LLMs. Notably, the smallest model, Qwen1.5-1.8B, achieves the best score. We conjecture this is due to it suffering the least stringent safety restrictions, thereby reducing instances of refusal to answer. Moreover, among the six core capabilities of MMStar, science &

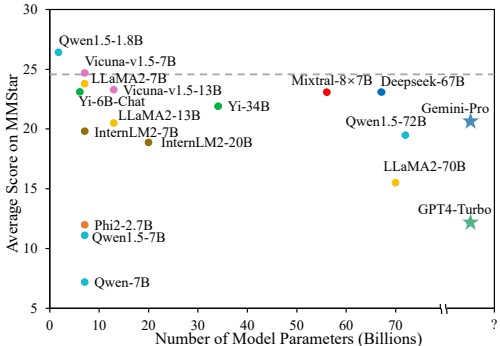

Figure 4: LLMs perform close to random guessing (the dashed line) on MMStar.

technology (ST) prove to be the most challenging dimension for LLMs. The best score on ST is only 23.2%, significantly lower than the best scores of around 30% in other dimensions. We speculate this may be that samples within the ST dimension have the least degree of data leakage from LLMs' training data.

**Observation from LVLMs.** We evaluate 2 closed-source and 14 open-source LVLMs on our MM-Star, with the results reported in Table 3. As shown in the table, GPT4V[43] with a high-resolution setting can achieve the best average score of 57.1% among all LVLMs. Increasing the resolution and number of image tokens can boost the average score from 46.1% to 57.1% for GPT4V, offering a positive signal to the research community. Among the open-source LVLMs, InternLM-Xcomposer2 [13] achieves an impressive score of 55.4%. LLaVA-Next [32] even surpasses GPT4V and GeminiPro-Vision [51] in the mathematics (MA) core capability. Notably, no LVLMs managed to reach a passing average score (*i.e.* $60\%$) in the core capabilities of fine-grained perception (FP), logical reasoning (LR), science & Technology (ST), and mathematics (MA), highlighting these dimensions as particularly challenging for existing LVLMs. Moreover, TinyLLaVA [69], despite its modest 3B scale, outperformed some competitors of 7B and even 13B surprisingly, underscoring the potential of smaller-scale LVLMs.

## 5.2 Analysis of Multi-modal Gain (MG) and Multi-modal Leakage (ML)

**Analysis from the model perspective.** In Table 4, we illustrate the MG/ML (Multi-modal Gain/Multi-modal Leakage) metrics for each LVLM across each benchmark and provide an average MG/ML metric across all benchmarks in the final column. For closed-source LVLMs, GPT4V demonstrates notable performance gains attributed to its multi-modal training, while GeminiPro-Vision shows lesser data leakage during multi-modal training. This suggests that GPT4V may have utilized a broader range of multi-modal training data compared to GeminiPro-Vision. Among the open-source LVLMs, InternLM-XComposer2 achieves the highest average multi-modal gain of 28.1 across all benchmarks, whereas LLaVA-1.5-7B records the lowest at 14.8. This outcome is reasonable given that LLaVA-1.5-7B employed the least amount of multi-modal training data among these open-source LVLMs. Despite LLaVA-1.5-7B having the lowest average multi-modal gain, it exhibits minimal multi-modal leakage. Additionally, models like Monkey-Chat, Sphinx-X-MoE, and Deepseek-VL display higher degrees of multi-modal leakage, highlighting the need for the community to consider this factor for fair comparisons.

**Analysis from the benchmark perspective.** In the final row of Table 4, we list the average multi-modal gain and multi-modal leakage for existing LVLMs across all benchmarks for analysis. MM-Bench registers the highest average multi-modal gain at 50.1, indicating a significant overlap between the domains covered by existing LVLMs' training data and MMBench. Conversely, MMMU exhibits the lowest average multi-modal gain at 5.8, suggesting a lesser degree of overlap between the domains of existing LVLMs' training corpora and those included in MMMU. Additionally, MM-Star, as expected, has the lowest degree of multi-modal leakage at 1.9. This provides a comprehen-

Table 4: **Evaluation of various LVLMs on 7 Benchmarks with multi-modal gain (MG) and multi-modal leakage (ML) metrics.** We report the results of 2 closed-source LVLMs and 14 open-source LVLMs with varying sizes and architectures. The bottom row represents the average across models for the same benchmark, while the rightmost column shows the average across benchmarks for the same LVLM. The **best** results are highlighted in **bold and underlined.** The *worst* results of MG and ML metrics are in *italic red*.

| Model | Param. | MMMU | | MMB | | ScienceQA | | AI2D | | SEED | | MathVista | | MMStar | | Avg. | |
|---|---|---|---|---|---|---|---|---|---|---|---|---|---|---|---|---|---|
| | | MG↑ | ML↓ | MG↑ | ML↓ | MG↑ | ML↓ | MG↑ | ML↓ | MG↑ | ML↓ | MG↑ | ML↓ | MG↑ | ML↓ | MG↑ | ML↓ |
| *Closed-source LVLMs* | | | | | | | | | | | | | | | | | |
| GPT4V[43] | - | **8.5** | 3.9 | **52.0** | 5.4 | 13.2 | 3.9 | 12.8 | 2.8 | **43.2** | 18.3 | **19.3** | **1.2** | **32.6** | 1.3 | **25.9** | 5.3 |
| GeminiPro-Vision[51] | - | 5.0 | **0.0** | 51.4 | **0.0** | **14.3** | **0.0** | **13.5** | **0.0** | 36.4 | **0.0** | 11.5 | **1.2** | 27.4 | **0.0** | 22.8 | **0.2** |
| *Open-source LVLMs* | | | | | | | | | | | | | | | | | |
| TinyLLaVA[69] | 3B | 6.0 | 10.0 | 45.9 | 13.8 | 6.8 | 15.2 | 10.5 | 13.2 | 32.9 | 10.8 | 5.4 | 1.5 | 16.4 | 7.6 | 17.7 | 10.3 |
| Yi-VL[62] | 6B | 5.3 | 7.4 | 45.6 | 14.1 | 5.1 | 9.4 | *3.9* | *16.6* | 29.2 | 10.9 | 3.8 | 3.0 | 15.6 | **0.0** | 15.5 | 8.8 |
| LLaVA-1.5[31] | 7B | 4.5 | **0.0** | *45.5* | 9.2 | 4.6 | 5.2 | 6.9 | 6.2 | 28.1 | 4.9 | 3.3 | **0.0** | *10.7* | **0.0** | *14.8* | 3.6 |
| ShareGPT4V[5] | 7B | 3.5 | 1.8 | 49.1 | 10.1 | 4.2 | 6.3 | 8.5 | 6.9 | 31.7 | 5.1 | 3.0 | 0.7 | 11.9 | **0.0** | 16.0 | 4.4 |
| InternLM-XC2[13] | 7B | 7.5 | 1.4 | 53.4 | *17.3* | **24.8** | 7.9 | **18.1** | 15.0 | 36.8 | 6.2 | **28.0** | *10.5* | 28.1 | 7.5 | **28.1** | 9.4 |
| Deepseek-VL[36] | 8B | 3.2 | 10.6 | 49.6 | 15.5 | 14.3 | 10.8 | 11.6 | 14.9 | 33.7 | 23.1 | 11.4 | 3.3 | 15.7 | **0.0** | 19.9 | 11.2 |
| Qwen-VL-Chat[2] | 10B | **10.0** | 4.2 | 49.6 | **0.3** | 11.0 | 4.0 | 12.3 | 6.4 | **44.5** | 11.9 | 11.4 | 0.3 | 23.9 | **0.0** | 23.2 | 3.9 |
| Monkey-Chat[30] | 10B | 4.7 | 12.6 | 55.4 | 7.2 | 11.3 | *18.4* | 11.7 | 14.2 | 33.0 | *28.5* | 9.0 | 4.5 | 13.5 | *11.1* | 19.8 | *13.8* |
| LLaVA-1.5[31] | 13B | 9.6 | **0.0** | 47.2 | 9.8 | 5.7 | 7.0 | 8.6 | 7.2 | 31.1 | 10.7 | 5.3 | 1.5 | 13.9 | **0.0** | 17.3 | 5.2 |
| CogVLM-Chat[55] | 17B | 4.1 | 0.2 | 47.9 | 5.2 | 11.7 | **0.0** | 10.8 | 10.0 | 32.0 | 4.1 | 9.7 | 3.0 | 14.9 | **0.0** | 18.7 | **3.2** |
| Yi-VL[62] | 34B | 5.9 | 0.2 | 48.3 | 12.7 | 6.7 | 15.0 | 6.0 | **2.6** | *27.1* | **3.7** | *2.9* | 1.0 | 18.8 | **0.0** | 16.5 | 5.0 |
| LLaVA-Next[32] | 34B | 6.6 | 2.8 | 54.7 | 4.8 | 11.2 | 1.5 | 12.8 | 5.6 | 34.1 | 6.7 | 16.5 | 4.3 | 29.4 | 2.4 | 23.6 | 4.0 |
| InternVL-Chat-v1.2[7] | 40B | 7.4 | 4.1 | **58.5** | 3.8 | 12.2 | 0.9 | 13.5 | 4.8 | 34.9 | 5.5 | 23.7 | 6.1 | **32.6** | **0.0** | 26.1 | 3.6 |
| Sphinx-X-MoE[17] | 57B | *1.2* | *17.9* | 48.7 | 11.9 | *3.8* | 11.2 | *3.9* | 12.4 | 31.2 | 26.4 | 9.7 | 5.0 | 14.8 | 1.0 | 16.2 | 12.3 |
| Avg. across models | - | *5.8* | 4.9 | **50.1** | 8.9 | 10.0 | 7.4 | 10.3 | 8.7 | 33.7 | *11.1* | 10.8 | 3.0 | 20.0 | **1.9** | - | - |

sive and fair arena for comparing existing LVLMs. Moreover, we believe evaluating existing LVLMs to derive average ML metrics can also be helpful to the following works in examining newly developed multi-modal benchmarks.

# 6 Conclusion

In this work, we dig into current evaluation works for large vision-language models (LVLMs) and identify two primary issues: 1) visual content is unnecessary for many samples, and 2) unintentional data leakage exists in LLM and LVLM training. To address these issues, we develop an elite vision-dependent multi-modal benchmark named MMStar and propose two metrics to measure the data leakage and actual performance gain in LVLMs' multi-modal training. MMStar undergoes the manual review of each sample, covering 6 core capabilities and 18 detailed axes for an in-depth evaluation of LVLMs' multimodal capabilities. In our evaluation of 16 diverse LVLMs on MM-Star, even the best model scores under 60 on average. We also analyze the MG and ML metrics across 6 multimodal benchmarks and MMStar, providing valuable insights for the community on gathering multimodal training data and crafting new benchmarks. In the future, we plan to expand MMStar into a larger, online test set and explore dynamic evaluation methods to maintain sample visual dependency and reduce accidental data leakage into LLM's and LVLM's training corpora.

# 7 Acknowledgments

This work was supported by the Anhui Provincial Natural Science Foundation under Grant 2108085UD12. We acknowledge the partial support of the GPU cluster built by MCC Lab of Information Science and Technology Institution, USTC. This work was also partially supported by the Shanghai Artificial Intelligence Laboratory, the National Key R&D Program of China (2022ZD0160201), the Centre for Perceptual and Interactive Intelligence (CPII) Ltd under the Innovation and Technology Commission (ITC)'s InnoHK. Dahua Lin is a PI of CPII under the InnoHK.

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

# A Appendix

In the supplementary material, we provide more results and analysis and summarize them as follows:

- In Section A.1, we detail the experimental setups.
- In Section A.2, we provide a detailed explanation of our MMStar benchmark, which encompasses definitions of six core capabilities and eighteen detailed axes.
- In Section A.3, we provide the details about the manual check process to aggressively reduce the MMStar benchmark from 11,607 to 1,500 samples.
- In Section A.4, we present the comprehensive performance of 22 LLMs across various dimensions on our MMStar benchmark.
- In Section A.5, we present statistics about the data leaked from the existing public multimodal benchmarks to selected LVLMs' training corpus, along with some specific examples.
- In Section A.6, we showcase additional samples from existing benchmarks that either lack visual dependency or have been leaked into the training corpora of LLMs or LVLMs.
- In Section A.7, we showcase some samples in MMStar of each detailed axe.
- In Section A.8, we provide detailed performance results of 22 LLMs across 6 public multimodal benchmarks under the 0-shot evaluation strategy. Moreover, we place the complete performance of 16 LVLMs with/without accessing images across these multi-modal benchmarks.
- In Section A.9, we discuss the limitation and future work.

## A.1 Experimental Setups

**Evaluation models.** 1) **Baseline**: We utilize random choice to serve as the baseline, which randomly selects an option as the answer. 2) **Large Language Models**: We prepare two closed-source LLMs, GPT4 [42] and GeminiPro [51], and 20 popular open-source LLMs sizing from 1.8B to 72B for text-only evaluation, such as Qwen series [1], LLaMA2 series [53], Phi2 [40], Vicuna series [9], Deepseek series [3], InternLM2 series [52], Baichuan2 series [60], Yi series [62], Mistral series [22, 23]. Additionally, all the open-source LLMs we used are their Chat versions. and 3) **Large Vision-Language Models**: We prepare two closed-source LVLMs, GPT4V [43] and GeminiPro (Vision) [51], and 14 popular open-source LVLMs sizing from 3B to 57B, such as TinyLLaVA-3B [69], Yi-VL series [62], Qwen-VL-Chat [2], LLaVA-1.5 series [31], LLaVA-Next-34B [32], CogVLM-Chat-17B [55], InternVL-Chat-v1.2 [7], Sphinx-X-8x7B [17].

**Implementation details.** For evaluating LLMs on existing benchmarks, we employ both 0-shot and 2-shot strategies and will specify which is utilized when reporting results. For evaluating LLMs on MMStar, the 0-shot strategy yields poor scores, making comparisons difficult. Therefore, we exclusively utilize the 2-shot strategy to decrease the frequency of refusal to answer. Moreover, All LVLMs are evaluated utilizing the 0-shot strategy across all benchmarks to ensure a fair comparison. When evaluating LVLMs under the 'LVLM-text' setting (*i.e.* answer without the image), most LVLMs work well by simply removing the image tokens from their default input tokens. However, GeminiPro-Vision [51] and CogVLM-Chat [55] require the replacement of the original images with pure grey images to bypass image content input and operate correctly. Given that all questions are ensured to be converted into a multiple-choice format, we develop some heuristic matching rules to calculate accuracy, avoiding the cumbersome process of re-invoking GPT4 for answer extraction. Moreover, all experiments in this study are conducted within the same codebase modified from VLMEvalKit [11], and utilize NVIDIA A100 GPUs for non-API-based evaluation.

## A.2 Definitions of Core Capabilities and Detailed Axes

The core capabilities consist of two perception-related dimensions, Coarse Perception (CP) and Fine-grained Perception (FP), two reasoning-related dimensions, Instance Reasoning (IR) and Logical Reasoning (LR), and two knowledge-related dimensions, Science & Technology (ST) and Mathematics (MA). We detail the complete definitions as follows:

**Coarse Perception (CP).** This core dimension refers to the capability to understand and interpret the overarching characteristics and themes of an image without delving into the finer details. It

encompasses a broad, holistic view of the visual content, enabling the identification of 1) image style & quality; 2) image scene & topic; and 3) image emotion.

**Fine-grained Perception (FP).** This core dimension represents a sophisticated level of image understanding that focuses on the detailed and nuanced aspects of visual content. It involves a deep dive into the specifics of images: 1) attribute & celebrity recognition; 2) object location; and 3) object counting. This core dimension unveils the subtle intricacies that coarse perception might overlook.

**Instance Reasoning (IR).** It encapsulates a set of advanced cognitive capabilities focused on understanding and interpreting individual and collective object attributes and interrelations within an image. This process goes beyond mere recognition, delving into the analytical assessment of 1) single-instance attribute reasoning; 2) cross-instance attribute comparison; and 3) cross-instance relation reasoning. It is a critical component for systems requiring a deep semantic understanding of visual content, enabling nuanced interaction with and response to complex visual content.

**Logical Reasoning (LR).** This core dimension encompasses a sophisticated framework of cognitive processes designed to interpret, deduce, and infer conclusions from visual content through a structured approach to logic and reasoning. This multi-faceted capability marries the intuitive understanding of visual content with the structured rigor of logical deduction, enabling: 1) diagram reasoning; 2) code & sequence reasoning; and 3) common reasoning.

**Science & Technology (ST).** It consists of a comprehensive framework for the application and integration of knowledge across a broad spectrum of science and technology. This domain combines the theoretical underpinnings and practical applications of various fields: 1) natural science; 2) engineering; and 3) geography & earth science.

**Mathematics (MA).** Math is a foundational pillar of logical and analytical reasoning and encompasses a broad spectrum of capabilities essential for understanding, applying, and interpreting quantitative and spatial information. We primarily consider three aspects for evaluating LVLMs' logical thinking prowess: 1) numeric commonsense & calculation; 2) geometry; and 3) statistical analysis.

### A.3 Details of Manual Check

After roughly filtering the original data pool with 8 advanced LLMs, resulting in 11,607 candidate samples, we initiate a rigorous manual review phase. First, we establish 6 core evaluation dimensions and 18 detailed axes by integrating the evaluation dimensions from existing benchmarks. Next, we use 16 LVLMs to infer and count the number of hits for each sample. Furthermore, we design a UI interface listing the current sample's image, options, answer, sample source, hit count, and the 18 detailed axes. The samples are arranged in ascending order based on the number of hits. The formal manual selection and benchmark construction process is as follows:

**Preliminary Classification:** Three experts are each responsible for two core capability dimensions (i.e., 6 detailed axes). They need to review all candidate samples and select and correctly classify the samples belonging to their respective dimensions. The samples selected must retain their visual dependency. Statistical Analysis: After the preliminary classification, we consider the numerical balance between dimensions and the difficulty level of the samples. Samples under the "coarse perception" dimension approach 4,000, while those under "logical reasoning" are fewer than 700. In terms of difficulty distribution, there are 4,555 easy (i.e., number of hits between 12 and 16) samples but only 2,758 tough (i.e., number of hits between 0 and 3) ones. Given these premises, a lot of repetitive simple samples, such as those merely asking for the color of an object in the image, are not what we desire.

**Initial Benchmark:** After considering both the numerical balance and difficulty level of the samples, we set the total sample number of the benchmark at 1,500, with each core capability dimension containing 250 samples. Then, we assign each expert two core capability dimensions, instructing them to prioritize sample difficulty when selecting 250 samples per dimension.

**Cross-Validation:** To minimize personal bias, we arrange for each expert to review the dimensions handled by the other two experts after the initial benchmark is constructed. Samples with issues are replaced by correct samples of the same difficulty level from the candidate pool. Moreover, we also provide the number of samples with consensus before and after the cross-validation step in the manual review process for MMStar in the table below. Only samples that all three experts unanimously

agree upon are retained; otherwise, they are replaced with samples of the same difficulty level from the candidate pool.

## A.4  Performance Comparison of Various LLMs on MMStar

Table 5: **LLMs failed to solve problems in MMStar and performed close to random guessing, visual content is necessary to solve MMStar.** We evaluate various LLMs on MMStar with the 2-shot inference strategy. We report the results of 2 closed-source LLMs and 20 open-source LLMs with varying sizes and architectures. We report the detailed results of the CP (coarse perception), FP (fine-grained perception), IR(instance reasoning), LR (logical reasoning), ST (science & technology), and MA (mathematics) core capabilities. The **best** results are highlighted in **bold and underlined.**

| Model | CP | FP | IR | LR | ST | MA | Avg. |
|---|---|---|---|---|---|---|---|
| *Baselines* | | | | | | | |
| Random Choice | 23.7 | 24.5 | 25.3 | 24.3 | 24.8 | 25.1 | 24.6 |
| *Closed-source LLMs* | | | | | | | |
| GPT4-Turbo[42] | 2.4 | 4.0 | 9.6 | 18.0 | 13.6 | 25.6 | 12.2 |
| Gemini-Pro[51] | **16.8** | **13.6** | **20.4** | **24.4** | **19.6** | **28.8** | **20.6** |
| *Open-source LLMs* | | | | | | | |
| Qwen1.5-1.8B[1] | 28.4 | 28.4 | 25.6 | 23.2 | **23.2** | 29.6 | **26.4** |
| Phi2-2.7B[40] | 11.2 | 11.2 | 15.2 | 10.8 | 11.6 | 12.0 | 12.0 |
| Yi-6B-Chat[62] | 23.6 | 19.2 | 28.4 | 25.2 | 12.4 | 29.6 | 23.1 |
| LLaMA2-7B[53] | 28.0 | **30.4** | 26.0 | 18.0 | 18.8 | 21.6 | 23.8 |
| Qwen-7B[1] | 11.6 | 5.6 | 12.8 | 5.6 | 7.2 | 0.4 | 7.2 |
| Deepseek-7B[3] | 26.8 | 16.0 | 28.4 | 21.6 | **23.2** | 25.6 | 23.6 |
| InternLM2-7B[52] | 22.0 | 14.8 | 22.0 | 21.6 | 15.2 | 23.2 | 19.8 |
| Qwen1.5-7B[1] | 15.6 | 8.0 | 9.2 | 9.2 | 15.2 | 9.2 | 11.1 |
| Vicuna-v1.5-7B[9] | 22.0 | 27.6 | 29.6 | 26.4 | 18.0 | 24.4 | 24.7 |
| Baichuan2-7B[60] | 20.8 | 18.4 | 27.6 | 18.8 | 18.8 | 21.2 | 20.9 |
| Mistral-7B[22] | 20.0 | 23.6 | 24.4 | 23.6 | 20.0 | 27.2 | 23.1 |
| LLaMA2-13B[53] | 23.6 | 23.6 | 28.0 | 21.2 | 16.4 | 10.4 | 20.5 |
| Vicuna-v1.5-13B[9] | **32.8** | 24.0 | **28.8** | 17.6 | 22.0 | 14.4 | 23.3 |
| Baichuan2-13B[60] | 26.4 | 18.0 | 28.0 | 20.4 | 21.2 | 25.6 | 23.3 |
| InternLM2-20B[52] | 18.2 | 17.8 | 22.6 | 23.8 | 17.8 | 13.4 | 18.9 |
| Yi-34B[62] | 20.4 | 18.0 | 24.0 | 24.0 | 14.4 | **30.8** | 21.9 |
| Mixtral-8x7B[23] | 24.4 | 17.6 | 19.2 | **28.0** | 16.0 | 33.6 | 23.1 |
| Deepseek-67B[3] | 29.2 | 22.4 | 18.4 | 26.0 | 20.4 | 22.4 | 23.1 |
| LLaMA2-70B[53] | 22.4 | 20.0 | 19.6 | 14.4 | 7.2 | 9.6 | 15.5 |
| Qwen1.5-72B[1] | 21.6 | 16.0 | 21.2 | 14.0 | 17.2 | 27.2 | 19.5 |

## A.5 Multi-modal Leakage in Existing Multi-modal Benchmarks

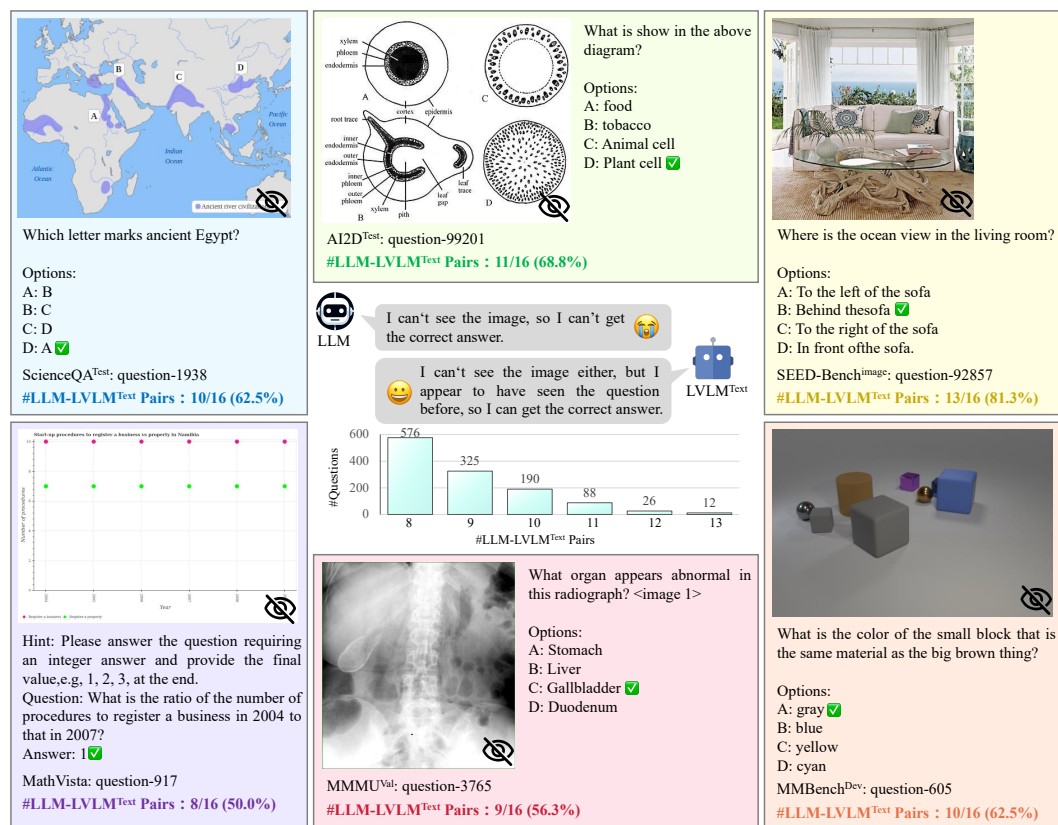

Figure 5: **Illustration of data leakage during LVLMs' multi-modal training processes.** We showcase samples that LLMs cannot answer correctly but LVLMs without accessing images (LVLM-text) can. Each LLM-LVLM$^{text}$ pair represents an LLM and its corresponding LVLM without accessing images, totaling 16 pairs. The chart in the center tallies the number of samples in existing benchmarks hit by more than half of the LLM-LVLM$^{text}$ pairs, underscoring the issue of data leakage during the multi-modal training process.

## A.6    More Failure Examples in Existing Multi-modal Benchmarks

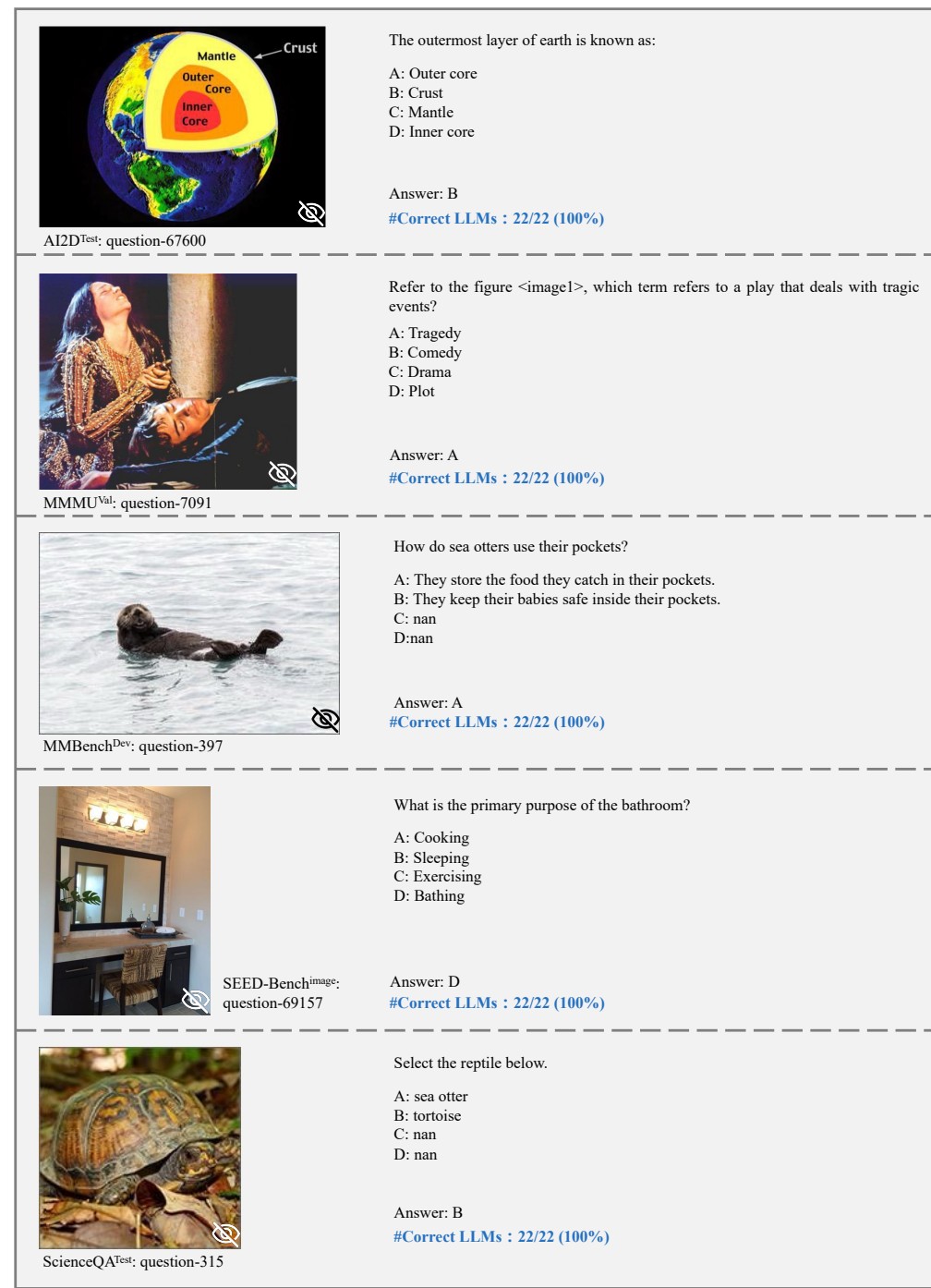

Figure 6: We highlight cases in existing benchmarks where evaluation samples lack the visual necessary.

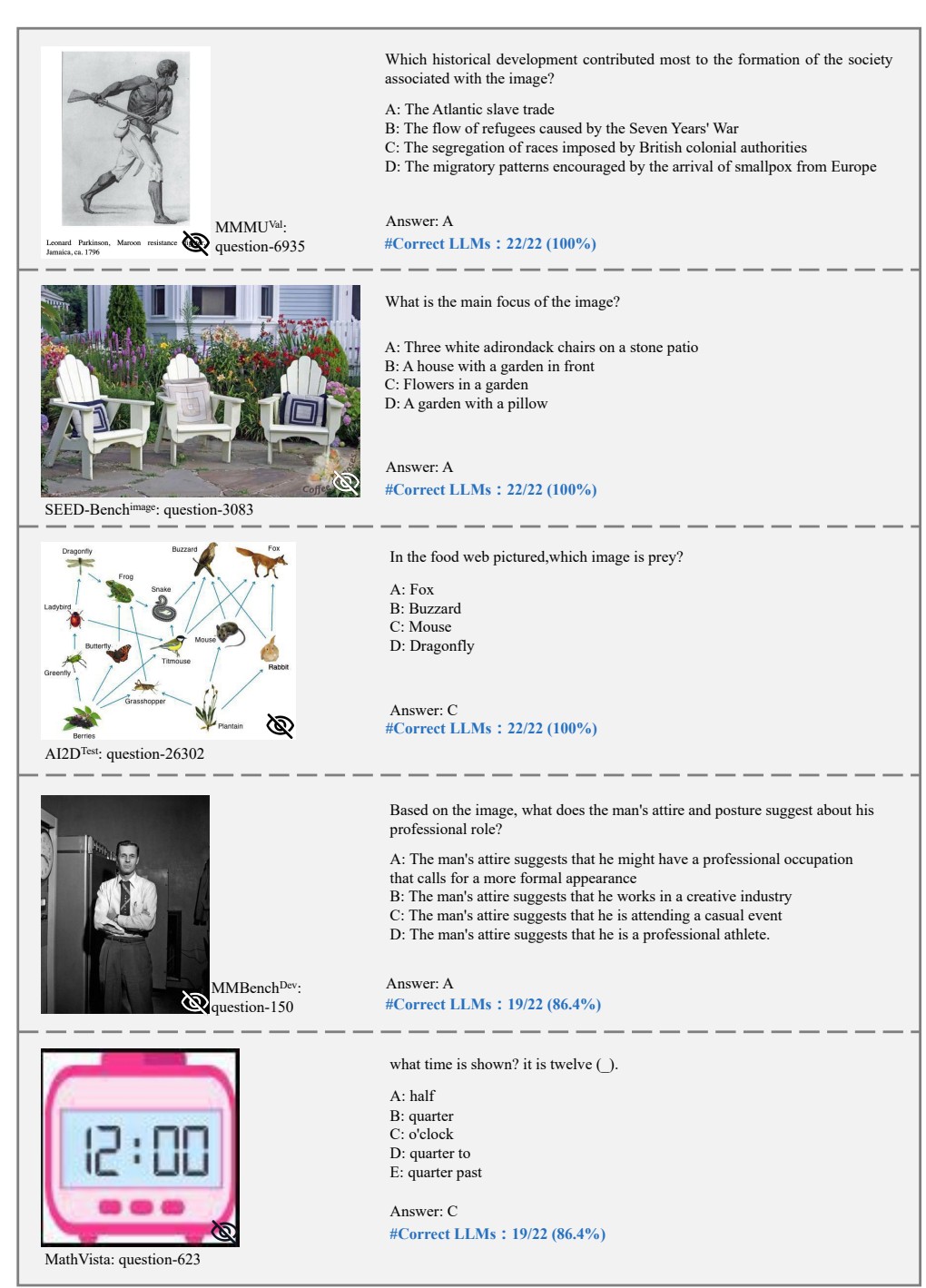

Figure 7: We highlight cases in existing benchmarks where evaluation samples are leaked into LLMs' training data.

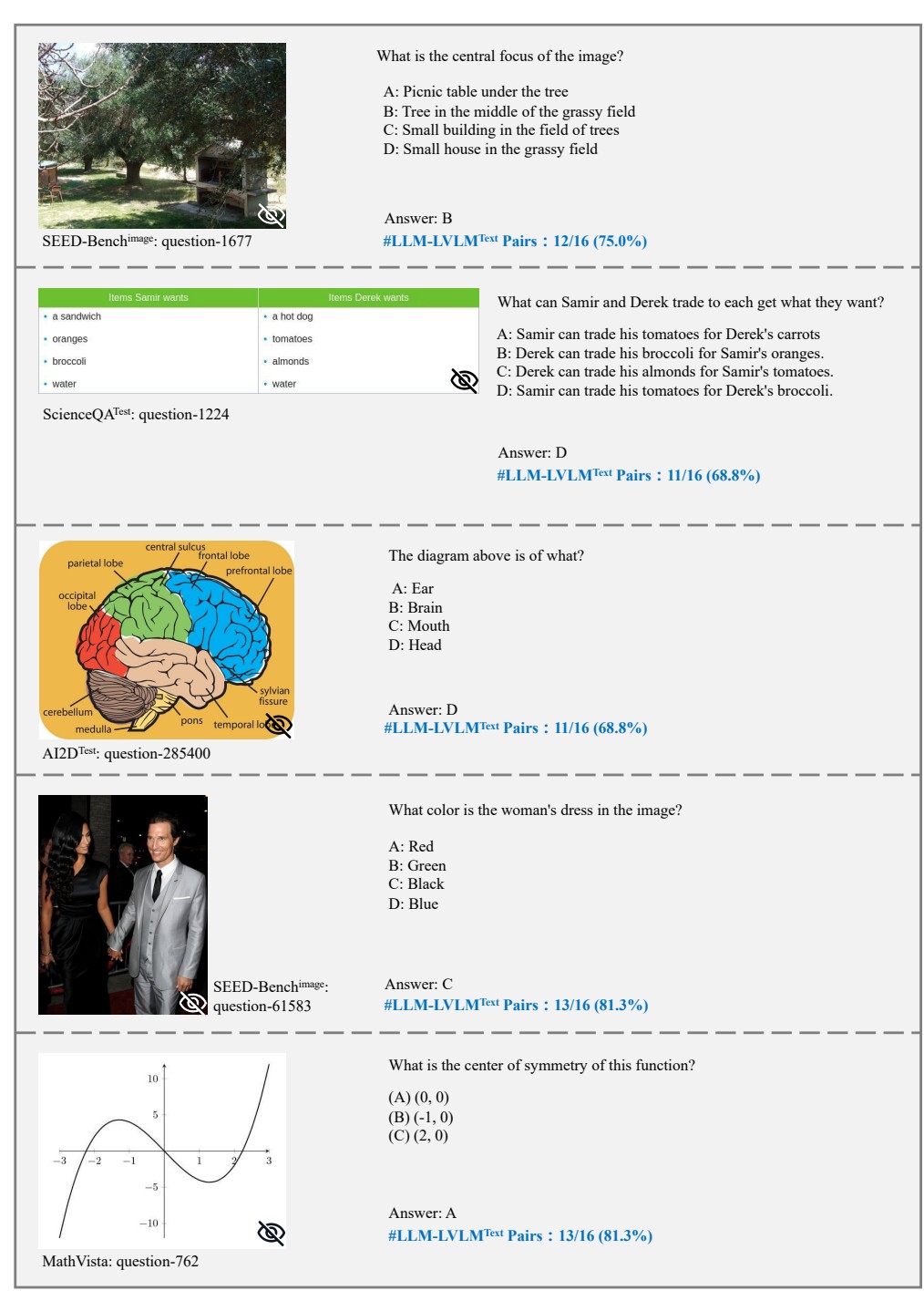

What is the central focus of the image?

A: Picnic table under the tree
B: Tree in the middle of the grassy field
C: Small building in the field of trees
D: Small house in the grassy field

Answer: B
#LLM-LVLMText Pairs : 12/16 (75.0%)

SEED-Benchimage: question-1677

| Items Samir wants | Items Derek wants |
|---|---|
| • a sandwich | • a hot dog |
| • oranges | • tomatoes |
| • broccoli | • almonds |
| • water | • water |

ScienceQATest: question-1224

What can Samir and Derek trade to each get what they want?

A: Samir can trade his tomatoes for Derek's carrots
B: Derek can trade his broccoli for Samir's oranges.
C: Derek can trade his almonds for Samir's tomatoes.
D: Samir can trade his tomatoes for Derek's broccoli.

Answer: D
#LLM-LVLMText Pairs : 11/16 (68.8%)

The diagram above is of what?

A: Ear
B: Brain
C: Mouth
D: Head

Answer: D
#LLM-LVLMText Pairs : 11/16 (68.8%)

AI2DTest: question-285400

What color is the woman's dress in the image?

A: Red
B: Green
C: Black
D: Blue

Answer: C
#LLM-LVLMText Pairs : 13/16 (81.3%)

SEED-Benchimage: question-61583

What is the center of symmetry of this function?

(A) (0, 0)
(B) (-1, 0)
(C) (2, 0)

Answer: A
#LLM-LVLMText Pairs : 13/16 (81.3%)

MathVista: question-762

Figure 8: We highlight cases in existing benchmarks where evaluation samples are leaked into LVLMs' multi-modal training data.

## A.7 More Examples in MMStar

### Coarse Perception

**image scene and topic**

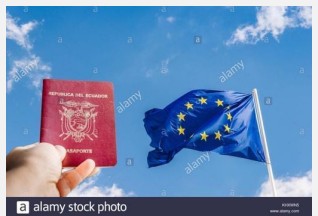

What is the predominant color in the image?
A: White,      B: Red,
C: Blue,      D: Silver

**image emotion**

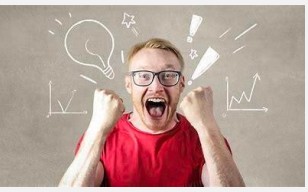

Which mood does this image convey?
A: Cozy,      B: Anxious,
C: Happy,      D: Angry

**image style & quality**

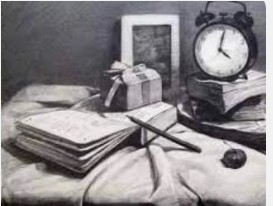

Which category does this image belong to?
A: oil painting,      B: sketch,
C: digital art,      D: photo

### Fine-grained Perception

**localization**

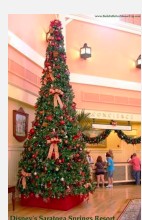

Where is the Christmas tree located in the image?
A: It is on the left-hand side of the image,
B: It is on the right-hand side of the image,
C: It is in the center of the image,
D: It is not in the image

**recognition**

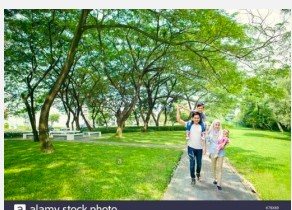

What is the main color of the shirt the woman is wearing?
A: White,      B: Blue,
C: Pink,      D: Black

**object counting**

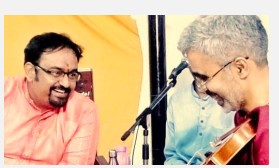

How many people are present in the image?
A: One,      B: Three,
C: Two,      D: Four

### Instance Reasoning

**single-instance reasoning**

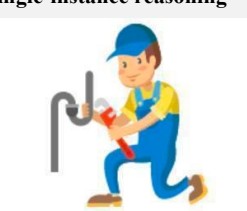

What's the profession of the people in this picture?
A: mason,      B: plumber,
C: pilot,      D: police

**cross-instance attribute reasoning**

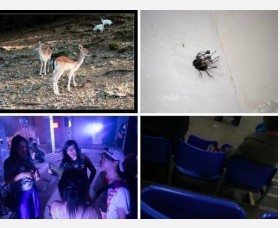

Which image is the brightest one?
A: upper left,      B: upper right,
C: down left,      D: down right

**cross-instance relation reasoning**

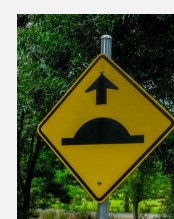

What is the relation between the arrow and the curve sign?
A: The arrow is pointing away from the curve sign,
B: The arrow is pointing to the curve sign,
C: The arrow and the curve sign are unrelated,
D: The arrow and the curve sign are overlapping

Figure 9: More examples in MMStar

## Logical Reasoning

**common reasoning**

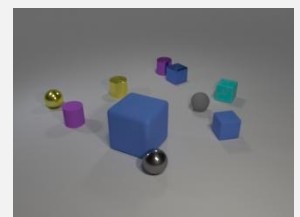

Subtract all yellow metallic balls. Subtract all small yellow shiny things. How many objects are left?
A: 4,          B: 5,
C: 6,          D: 8 ✓

**diagram reasoning**

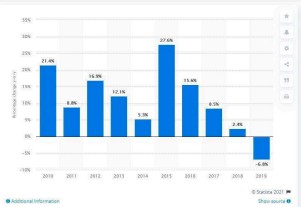

How many years have value less than 10%?
A: 0,          B: 1,
C: 2,          D: 5 ✓

**code & sequence reasoning**

```
The count is: 0
The count is: 1
The count is: 2
The count is: 3
The count is: 4
The count is: 5
The count is: 6
The count is: 7
The count is: 8
Good bye!
```

Which Python code can generate the content of the image?
A: count = 0 while (count < 10): print 'The count is:', count count = count + 1 print "Good bye!",
B: count = 0 while (count < 9): print 'The count is:', count count = count + 1 print "Good bye!",
C: count = 1 while (count < 9): print 'The count is:', count count = count + 1 print "Good bye!",
D: count = 0 while (count < 9): print 'The count is:', count count = count + 2 print "Good bye!"

## Science & Technology

**biology & chemistry & physics**

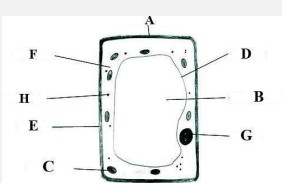

what is label e in diagram?
A: nucleus,          B: chloroplast,
C: cell wall, ✓     D: cell sap vacuole

**geography & earth science & agriculture**

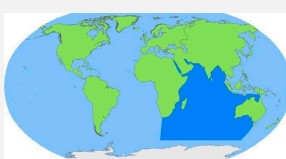

Which ocean is highlighted?
A: the Indian Ocean, ✓
B: the Atlantic Ocean,
C: the Pacific Ocean,
D: the Southern Ocean

**electronics & energy & mechanical eng.**

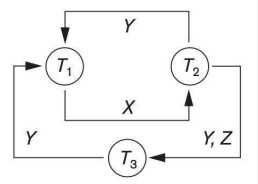

Which schedule is an equivalent serial schedule for the precendence graph in <image 1>? A: T3 -> T1 -> T2,
B: T2 -> T1 -> T3,
C: T1 -> T2 -> T3,
D: There are no serial schedules for the graph. ✓

## Mathematics

**geometry**

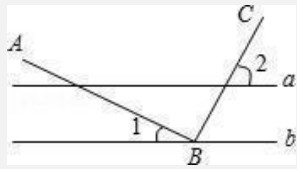

As shown in the figure, the straight line a ∥ b, the point B is on the straight line b, and AB ⊥ BC, ∠2 = 65.0, then the degree of ∠1 is ()
A 65°          B 25° ✓
C 35°          D 45°

**statistical reasoning**

| Year | Inflation, % | Stock Market Return, % | T-Bill Return, % |
|------|------|------|------|
| 1929 | −0.2 | −14.5 | 4.8 |
| 1930 | −6.0 | −28.3 | 2.4 |
| 1931 | −9.5 | −43.9 | 1.1 |
| 1932 | −10.3 | −9.9 | 1.0 |
| 1933 | 0.5 | 57.3 | 0.3 |

What was the real return on the stock market in 1932?
A: -14.33%,          B: -23.72%,
C: 0.45%, ✓         D: 56.52%

**numeric commonsense and calculation**

| | |
|---|---|
| spelt rolls | $8/kilogram |
| hamburger buns | $7/kilogram |
| rye rolls | $6/kilogram |
| wheat rolls | $8/kilogram |
| English muffins | $8/kilogram |
| tortillas | $5/kilogram |

Colton wants to buy 1+3/10 kilograms of English muffins. How much will he spend? (Unit: $)
A 10.4 ✓          B 5.2
C 0               D 1

Figure 10: More examples in MMStar

## A.8 More Results on Public Multi-modal Benchmarks

Table 6: **Evaluation of various LLMs on six popular multi-modal benchmarks.** We employ a 0-shot inference strategy for evaluating all LLMs. We report the results of 2 closed-source LLMs and 20 open-source LLMs with varying sizes and architectures. The evaluated benchmarks include MMMU (MMMU-Val [64]), MMB (MMBench-EN-Dev [34]), ScienceQA (ScienceQA-Test [38]), AI2D (AI2D-Test [26]), SEED (SEED-Image [27]), and MathVista (MathVista-Mini [37]). The **best** results are highlighted in **bold and underlined.**

| Model | Strategy | MMMU | MMB | ScienceQA | AI2D | SEED | MathVista | Avg. |
|---|---|---|---|---|---|---|---|---|
| *Baselines* | | | | | | | | |
| Random Choice | - | 22.1 | 0.0 | 24.2 | 23.8 | 24.3 | 17.9 | 18.7 |
| *Closed-source LLMs* | | | | | | | | |
| GPT4-Turbo[42] | 0-shot | 41.2 | 12.2 | 64.3 | **59.7** | 10.1 | **24.2** | 35.3 |
| GeminiPro[51] | 0-shot | **42.9** | **18.4** | **68.9** | 59.2 | **35.5** | 23.3 | **41.4** |
| *Open-source LLMs* | | | | | | | | |
| Qwen1.5-1.8B[1] | 0-shot | 29.0 | 10.0 | 54.3 | 37.9 | 28.9 | 20.4 | 30.1 |
| Phi2-2.7B[40] | 0-shot | 20.0 | 7.2 | 47.1 | 38.7 | 26.4 | 22.0 | 26.9 |
| Yi-6B[62] | 0-shot | 25.7 | 9.5 | 58.1 | 39.1 | 27.4 | 21.2 | 30.2 |
| LLaMA2-7B[53] | 0-shot | 23.6 | 11.5 | 56.8 | 43.5 | 31.7 | 24.1 | 31.9 |
| Qwen-7B[1] | 0-shot | 19.8 | 8.4 | 52.7 | 42.6 | 7.6 | 20.5 | 25.3 |
| Deepseek-7B[3] | 0-shot | 21.6 | 8.4 | 56.3 | 38.1 | 13.4 | 20.6 | 26.4 |
| InternLM2-7B[52] | 0-shot | 32.8 | 8.9 | 64.0 | 48.3 | 31.9 | 18.9 | 34.1 |
| Qwen1.5-7B[1] | 0-shot | 25.0 | 11.4 | 62.3 | 49.4 | 19.4 | 19.9 | 31.2 |
| Vicuna-v1.5-7B[9] | 0-shot | 29.9 | 10.3 | 58.9 | 42.5 | 32.6 | 22.0 | 32.7 |
| Baichuan2-7B[60] | 0-shot | 25.7 | 10.5 | 52.7 | 44.0 | 29.2 | 20.8 | 30.5 |
| Mistral-7B[22] | 0-shot | 30.0 | 13.2 | 63.4 | 48.5 | 34.3 | 22.6 | 35.3 |
| LLaMA2-13B[53] | 0-shot | 24.4 | 10.1 | 59.1 | 45.0 | 33.6 | 23.8 | 32.7 |
| Vicuna-v1.5-13B[9] | 0-shot | 28.3 | 11.6 | 59.5 | 45.0 | 26.3 | 19.6 | 31.7 |
| Baichuan2-13B[60] | 0-shot | 22.1 | 4.7 | 51.1 | 32.8 | 25.4 | 20.3 | 26.1 |
| InternLM2-20B[52] | 0-shot | 32.2 | **15.9** | 63.8 | 55.7 | 26.0 | 21.3 | 35.8 |
| Yi-34B[62] | 0-shot | **37.1** | 10.5 | 53.6 | 57.3 | **37.3** | 21.7 | **36.3** |
| Mixtral-8x7B[23] | 0-shot | 25.7 | 8.6 | 57.2 | 48.7 | 13.5 | 23.4 | 29.5 |
| Deepseek-67B[3] | 0-shot | 30.9 | 14.8 | **64.3** | **57.5** | 17.1 | 23.2 | 34.6 |
| LLaMA2-70B[53] | 0-shot | 28.9 | 12.3 | 62.2 | 48.6 | 34.3 | **25.2** | 35.3 |
| Qwen1.5-72B[1] | 0-shot | 21.4 | 10.1 | 57.5 | 44.2 | 8.8 | 19.5 | 26.9 |

Table 7: **Evaluation of various LVLMs on six popular multi-modal benchmarks.** For the "strategy" column, "LLM" refers to evaluating using the corresponding LLM base of the LVLM, while "LVLM-text" denotes evaluating LVLMs without accessing images. We employ the 0-shot inference strategy for LLMs to align the evaluation protocols of LVLMs. The highest results of the LVLM-text setting across the models are highlighted in **bold and underlined.**

| Model | Param. | Strategy | MMMU | MMB | ScienceQA | AI2D | SEED | MathVista | Avg. |
|---|---|---|---|---|---|---|---|---|---|
| | | | *Baseline* | | | | | | |
| Random Choice | - | - | 22.1 | 0.0 | 24.2 | 23.8 | 24.3 | 17.9 | 18.7 |
| | | | *Closed-source LVLMs and corresponding LLM bases* | | | | | | |
| GPT4V[43] (GPT4-Turbo[42]) | - | LLM | 41.2 | 12.2 | 64.3 | 59.7 | 10.1 | 24.2 | 35.3 |
| | | LVLM-text | **45.1** | **17.6** | **68.2** | **62.5** | **28.4** | **25.4** | **41.2** |
| | | LVLM | 53.6 | 69.6 | 81.4 | 75.3 | 71.6 | 44.7 | 66.0 |
| GeminiPro-Vision[51] (GeminiPro[51]) | - | LLM | 42.9 | 18.4 | 68.9 | 59.2 | 35.5 | 23.3 | 41.4 |
| | | LVLM-text | 39.4 | 16.7 | 66.3 | 54.5 | 27.9 | 24.5 | 38.2 |
| | | LVLM | 44.4 | 68.1 | 80.6 | 68.0 | 64.3 | 36.0 | 60.2 |
| | | | *Open-source LVLMs and corresponding LLM bases* | | | | | | |
| TinyLLaVA[69] (Phi2-2.7B[40]) | 3B | LLM | 20.0 | 7.2 | 47.1 | 38.7 | 26.4 | 22.0 | 26.9 |
| | | LVLM-text | 30.0 | 21.0 | 62.3 | 51.9 | 37.2 | 23.5 | 37.7 |
| | | LVLM | 36.0 | 66.9 | 69.1 | 62.4 | 70.1 | 28.9 | 55.6 |
| Yi-VL[62] (Yi-6B[62]) | 6B | LLM | 25.7 | 9.5 | 58.1 | 39.1 | 27.4 | 21.2 | 30.2 |
| | | LVLM-text | 33.1 | 23.6 | 67.5 | 55.7 | 38.3 | 24.2 | 40.4 |
| | | LVLM | 38.4 | 69.2 | 72.6 | 59.6 | 67.5 | 28.0 | 55.9 |
| LLaVA-1.5[31] (Vicuna-v1.5-7B[9]) | 7B | LLM | 29.9 | 10.3 | 58.9 | 42.5 | 32.6 | 22.0 | 32.7 |
| | | LVLM-text | 29.9 | 19.5 | 64.1 | 48.7 | 37.5 | 20.3 | 36.7 |
| | | LVLM | 34.4 | 65.0 | 68.7 | 55.6 | 65.6 | 23.6 | 52.2 |
| ShareGPT4V[5] (Vicuna-v1.5-7B[9]) | 7B | LLM | 29.9 | 10.3 | 58.9 | 42.5 | 32.6 | 22.0 | 32.7 |
| | | LVLM-text | 31.7 | 20.4 | 65.2 | 49.4 | 37.7 | 22.7 | 37.9 |
| | | LVLM | 35.2 | 69.5 | 69.4 | 57.9 | 69.4 | 25.7 | 54.5 |
| InternLM2-XC2[13] (InternLM2-7B[52]) | 7B | LLM | 32.8 | 8.9 | 64.0 | 48.3 | 31.9 | 18.9 | 34.1 |
| | | LVLM-text | 34.2 | **26.2** | **71.9** | 63.3 | 38.1 | **29.4** | 43.9 |
| | | LVLM | 41.7 | 79.6 | 96.7 | 81.4 | 74.9 | 57.4 | 72.0 |
| Qwen-VL-Chat[2] (Qwen-7B[1]) | 8B | LLM | 19.8 | 8.4 | 52.7 | 42.6 | 7.6 | 20.5 | 25.3 |
| | | LVLM-text | 24.0 | 8.7 | 56.7 | 49.0 | 19.5 | 20.8 | 29.8 |
| | | LVLM | 34.0 | 58.3 | 67.7 | 61.3 | 64.0 | 32.2 | 52.9 |
| Deepseek-VL[36] (Deepseek-7B[3]) | 8B | LLM | 21.6 | 8.4 | 56.3 | 38.1 | 13.4 | 20.6 | 26.4 |
| | | LVLM-text | 32.2 | 23.9 | 67.1 | 53.0 | 36.5 | 23.9 | 39.4 |
| | | LVLM | 35.4 | 73.5 | 81.4 | 64.6 | 70.2 | 35.3 | 60.1 |
| Monkey-Chat[30] (Qwen-7B[1]) | 10B | LLM | 19.8 | 8.4 | 52.7 | 42.6 | 7.6 | 20.5 | 25.3 |
| | | LVLM-text | 32.4 | 15.6 | 71.1 | 56.8 | 36.1 | 25.0 | 39.5 |
| | | LVLM | 37.1 | 71.0 | 82.4 | 68.5 | 69.1 | 34.0 | 60.4 |
| LLaVA-1.5[31] (Vicuna-v1.5-13B[9]) | 13B | LLM | 28.3 | 11.6 | 59.5 | 45.0 | 26.3 | 19.6 | 31.7 |
| | | LVLM-text | 26.0 | 21.4 | 66.5 | 52.2 | 37.0 | 21.1 | 37.4 |
| | | LVLM | 35.6 | 68.6 | 72.2 | 60.8 | 68.1 | 26.4 | 55.3 |
| CogVLM-Chat[55] (Vicuna-v1.5-7B[9]) | 17B | LLM | 29.9 | 10.3 | 58.9 | 42.5 | 32.6 | 22.0 | 32.7 |
| | | LVLM-text | 30.1 | 15.5 | 54.6 | 52.5 | 36.7 | 25.0 | 35.7 |
| | | LVLM | 34.2 | 63.4 | 66.3 | 63.3 | 68.7 | 34.7 | 55.1 |
| Yi-VL[62] (Yi-34B[62]) | 34B | LLM | 37.1 | 10.5 | 53.6 | 57.3 | 37.3 | 21.7 | 36.3 |
| | | LVLM-text | 37.3 | 23.2 | 68.6 | 59.9 | **41.0** | 22.7 | 42.1 |
| | | LVLM | 43.2 | 71.5 | 75.3 | 65.9 | 68.1 | 25.6 | 58.3 |
| LLaVA-Next[32] (NH2-Yi-34B[41]) | 34B | LLM | 37.6 | 20.1 | 69.4 | 60.2 | 35.0 | 17.9 | 37.2 |
| | | LVLM-text | 40.4 | 24.9 | 70.9 | 65.8 | 41.7 | 22.2 | 44.3 |
| | | LVLM | 47.0 | 79.6 | 82.1 | 78.6 | 75.8 | 38.7 | 67.0 |
| InternVL-Chat-v1.2[7] (NH2-Yi-34B[41]) | 40B | LLM | 37.6 | 20.1 | 69.4 | 60.2 | 35.0 | 17.9 | 40.0 |
| | | LVLM-text | 41.7 | 23.9 | 70.3 | **65.0** | 40.5 | 24.0 | **44.2** |
| | | LVLM | 49.1 | 82.4 | 82.5 | 78.5 | 75.4 | 47.7 | 69.3 |
| Sphinx-X-MoE[17] (Mixtral-8x7B[23]) | 57B | LLM | 25.7 | 8.6 | 57.2 | 48.7 | 13.5 | 23.4 | 29.5 |
| | | LVLM-text | **43.6** | 20.5 | 68.4 | 61.1 | 39.9 | 28.4 | 43.7 |
| | | LVLM | 44.8 | 69.2 | 72.2 | 65.0 | 71.1 | 38.1 | 60.1 |

## A.9 Limitations

While we have expended significant effort to filter out evaluation samples that are visually dependent and have not been leaked into the training corpora of existing LLMs and LVLMs for our MMStar benchmark, it is challenging to ensure that these samples will not be inadvertently included in the expanded training materials of future LLMs and LVLMs. Although the metrics we proposed, such as multi-modal gain and multi-modal leakage, can reflect this issue to some extent, a test set without provided answers is still needed to further assess the actual multi-modal capabilities of existing LVLMs. We plan to construct a new set of visual-dependent test samples for MMStar-Test in our future work.

