# OpenReview forum: "Are We on the Right Way for Evaluating Large Vision-Language Models?"
_NeurIPS.cc/2024/Conference — NeurIPS 2024 poster_

### Official Review · Reviewer_sNEM · 2024-06-13

**Soundness:** 2
**Presentation:** 3
**Contribution:** 2
**Rating:** 6
**Confidence:** 4

**Summary:**

This paper investigates the evaluation of large vision-language models (LVLMs) and the currently used benchmarks. Within the paper two primary issues are identified: the lack of need for visual information, and data leakage. Based on these issues a new compiled benchmark is proposed MMStar that includes a set of multi-modal samples validated by humans, and a variety of models are evaluated on this new benchmark.

**Strengths:**

With the rapid progress of the field of LLM and LVLMs, it is crucial that we have reliable benchmarks for evaluation - this work explores existing benchmarks, identifies potential issues therein, and proposes a new benchmark which contributes to the quality of evaluation in the field. Additionally, the work proposes new metrics for evaluation and performs a benchmark evaluation of various models.

**Weaknesses:**

1) Random Chance

An issue in this work is that it insufficiently accounts for random guessing and data bias. As shown in [27], the ground-truth chance for it being answer A or B on MMbench are 26.4% (due to having questions with less than 4 options), and moreover, certain models may prefer certain answers. LLMs may get correct answers even when they require visual input. Given the 26.4% baseline for MMbench from [27], it is also surprising that the random choice value reported in this paper, for MMbench, is 0.0 in both Table 1 and Table 2. Moreover, an additional baseline based on majority class may also be necessary here.

For the results in Figure 2, which require 6 out of 8 models to get a hit it is unlikely that random guessing has a big influence. However, for the results in Table 1 this is not the case. For instance, it appears that all MMBench results in Table 1 are below the 26.4% from [27] - which means all LLM do worse than always answering A (or B). For other benchmarks there may similarly be data biases in which answer is more frequent.

Relatedly, it may be the case that such data biases about which options are more frequently chosen are more pronounced for multi-modal questions, i.e., when asked about colour the answer is more often grey across all datasets and settings - which means LLM do not learn this data bias as their training data does not include such questions, but LVLM may learn this because the same bias is present in their training data. While this is unlikely to fully explain the phenomenon observed in Table 2, it may explain part of it and not be directly related to data leakage.

2) Not all benchmarks

The issues identified with existing benchmarks do not hold evenly across the benchmarks tested. In particular, MMBench and slightly lesser, MathVista seem to do pretty well with respect to these issues. This also appears to be reflected in the construction of MMStar, where after Manual Review the proportion of questions from these two existing benchmarks jumps considerable - in the end making up almost half of the MMStar benchmark. Which raises questions whether there is any benefit of using MMStar versus simply using MathVista and MMBench.

3) Manual review

The manual review description is insufficiently clear. From the three criteria applied, only the first one is somewhat clear - for the other two it is unclear how the 'experts' judged this. I would expect description (can be in appendix) of agreement rates between these experts for these criteria, as well as further description of what these criteria entail.

4) New metrics

The newly proposed metrics MG and ML are somewhat unclear in what they measure. If the 1500 questions in MMStar all require visual input as determined by the manual review, then what additional information does the MG metric give? Given the discussion above about chance, it appears the MG metric is more of a correction for random guessing. The ML metric similarly doesn't account for random guessing, or the aforementioned potential for data biases.

**Questions:**

The paper raises a number of interesting questions and points out limitations of (some) existing benchmarks. Unfortunately, the data leakage issue appears to be not fully disentangled from the potential of data biases. I would appreciate input from the authors on this point.

Additionally, I would be interested in discussion on random guessing within the paper, and whether the ideas presented in [27] may address this - and then subsequently how this influences the findings in the paper. Also taking into account the majority class versus random guess.

**Limitations:**

I believe question 9 in the paper checklist has been answered incorrectly - or at least the justification given is not valid. Given that the paper discusses a benchmark that may include images of people or copyrighted material it is crucial that the authors affirm whether the work has been done in accordance to the ethical guidelines. Even if the data is a compilation of existing datasets - by selecting and combining information from these the new resulting dataset may be biased in ways the original datasets were not (e.g., by selecting only those images containing users from a certain demographic).

---

> ### Author Rebuttal · Authors · 2024-08-06
>
> Thanks for your valuable suggestions. We address your concerns point by point:
>
> **Q1: Given the 26.4% baseline for MMbench from [27], it is also surprising that the random choice value reported in this paper, for MMbench, is 0.0 in both Table 1 and Table 2.**
>
> **A1:** The 26.4% in MMBench represents the frequent choice probability. We use MMBench's circular evaluation method, where:
>
> 1. Options are shuffled and evaluated multiple times
> 2. All evaluations must be correct for the sample to be considered correct
>
> Probability of random guessing:
>
> - Two-choice: 1/4
> - Three-choice: 1/27
> - Four-choice: 1/256
>
> MMBench has 77 two-choice, 221 three-choice, and 1462 four-choice questions, resulting in a theoretical accuracy of approximately 1.88%. Using VLMEvalKit, the files include options C and D for all samples, making the generalized random choice accuracy about 0.39%. Actual random selection shows an accuracy of 0.01%, rounded to 0.0% in Tables 1 and 2. This reflects the presence of samples that do not rely on visual content or have been leaked in the LLM training data, aligning with our findings.
>
> **Q2: An additional baseline based on majority class may also be necessary here.**
>
> **A2:** Good point! We conducted a frequent choice evaluation on existing benchmarks and MMStar:
>
> - MMMU|26.8
> - MMB|0.0
> - ScienceQA|36.0
> - AI2D|26.9
> - SEED|26.9
> - MathVista|26.3
> - MMStar|29.8
>
> The result of 0.0% for MMBench is due to circular evaluation, too. We chose not to use the circular evaluation mechanism for MMStar, consistent with most other benchmarks. We'll include these results in the next version.
>
> **Q3: For the results in Figure 2, which require 6 out of 8 models...which means all LLM do worse than always answering A (or B).**
>
> **A3:** This misunderstanding is resolved by explaining the circular evaluation for MMBench. The 0.0% results for random choice and frequent choice are lower than the 13.8% average accuracy for 22 LLMs. This indicates some samples don't rely on visual content or have been leaked in LLM training data, aligning with our observations.
>
> **Q4: The issues identified with existing benchmarks...whether there is any benefit of using MMStar versus simply using MathVista and MMBench.**
>
> **A4:** On one hand, MMBench uses circular evaluation, so the 0.0% random choice accuracy is much lower than the 13.8% average for the 22 LLMs. On the other hand, MathVista's 17.9% random choice accuracy is also lower than the 22.5% average for the 22 LLMs. While MathVista focuses on mathematics, our MMStar covers six core competencies and 18 detailed axes, offering a more comprehensive evaluation of LVLMs' multimodal capabilities. Additionally, we manually verified that each MMStar sample is visually dependent, a guarantee that neither MMBench nor MathVista can provide.
>
> **Q5: The manual review description is insufficiently clear...as well as further description of what these criteria entail.**
>
> **A5:** Due to space constraints, we have detailed the manual review process and the agreement rates in the General Author Rebuttal.
>
> **Q6: The newly proposed metrics MG and ML...account for random guessing, or the aforementioned potential for data biases.**
>
> **A6:** The score of an LVLM on a multimodal benchmark can be split into three parts: leakage from LLMs, leakage during multimodal training, and genuine understanding from multimodal training. Multi-modal Leakage (ML) measures the second part, and Multi-modal Gain (MG) measures the third. These metrics complement each other and should not be considered separately.
>
> These metrics are not exclusive to MMStar; they can analyze existing benchmarks that may not ensure visual dependency. Although MMStar ensures visual dependency, some samples might still leak into future LVLMs' training corpora. In such cases, MG and ML can assess multi-modal training leakage and performance gains.
>
> **Q7: The paper raises a number of interesting questions...input from the authors on this point.**
>
> **A7:** We have provided clarifications in our previous responses to Q1, Q2, and Q3, explaining that we adopt MMBench's native circular evaluation mechanism. Therefore, the 0.0% accuracy for both random choice and frequent choice is significantly lower than the average accuracy of 13.8% for the 22 LLMs, which indicates the presence of data leakage in the LLMs.
>
> **Q8: Additionally, I would be interested...Also taking into account the majority class versus random guess.**
>
> **A8:** We add circular evaluation results for MMBench, AI2D, and MMStar using two representative LVLMs. One can observe from the table that:
>
> - Random choice and frequent choice results close to 0% under circular evaluation
> - Data leakage in LLMs observed (e.g., InternVL-Chat-v1.2's LLM achieves 47.3% on AI2D, surpassing LLaVA-1.5's performance with images)
> - Multimodal training data leakage evident (e.g., LLaVA-1.5 improves MMBench and AI2D performance by 8.8% and 14.3% without image input)
> - MMStar mitigates sample leakage in LLM and LVLM training corpora while maintaining the challenge level
>
> | Model              | Strategy  |  MMB | AI2D | MMStar |
> |--------------------|-----------|:----:|:----:|:------:|
> | Random Choice      | -         |  0.0 |  0.2 |   0.1  |
> | Frequent Choice    | -         |  0.0 |  0.0 |   0.0  |
> | LLaVA-1.5          | LLM       | 10.3 | 18.3 |   1.7  |
> |                    | LVLM-text | 19.5 | 32.6 |   6.7  |
> |                    | LVLM      | 65.0 | 41.7 |  19.0  |
> | InternVL-Chat-v1.2 | LLM       | 20.1 | 47.3 |   4.1  |
> |                    | LVLM-text | 23.9 | 53.3 |  11.1  |
> |                    | LVLM      | 82.4 | 71.7 |  45.6  |
>
> **Q9: I believe question 9 in the paper checklist has been answered incorrectly.**
>
> **A9:** We have carefully reviewed the Code of Ethics of NeurIPS and cross-checked each requirement to ensure compliance. We will update this answer to [Yes] in the next version of the manuscript.
>
> Please do not hesitate to contact us if you have any further questions.

---

> > ### Comment · Reviewer_sNEM · 2024-08-12
> >
> > Thank you for the detailed reply and clarifications, with this my concerns are addressed, I will update my score to a weak accept.

---

> > > ### Author Response · Authors · 2024-08-12
> > > **Official Comment by Authors**
> > >
> > > Dear Reviewer sNEM:
> > >
> > > Thank you for your time and patience in reviewing our submission and rebuttal. Your appreciation for our work, along with the questions and suggestions you provided, has greatly helped us improve the quality of our work.
> > >
> > > Best regards and thanks,
> > >
> > > Paper 2491 Authors

---

### Official Review · Reviewer_JQaJ · 2024-06-25

**Soundness:** 3
**Presentation:** 3
**Contribution:** 3
**Rating:** 7
**Confidence:** 4

**Summary:**

The current benchmarks used to evaluate Vision Language Models (VLMs) contain several flaws. In particular, a lot of questions can be answered without looking at the image at all. These benchmarks still being hard, the best proprietary models without looking at the images can obtain better scores than strong VLM baselines (looking at the images). As a result, the authors create MMStar, a difficult benchmark aiming at evaluating the capability of the vision-language tasks. They manually review their benchmark, and do several ablations to confirm its importance.

**Strengths:**

- The authors provide a benchmark that is hard to be good at without looking at the images. This is a problem for some questions in MMMU and MathVista currently.
- The authors manually reviewed the examples of the benchmark.
- The dataset is nicely divided into 6 subtasks, evaluating different aspects. The fact that there is exactly the same number of examples in each of these subtasks is appreciated.
- The fact that each question is a MCQ, instead of an open-ended question that would be difficult to evaluate due to the different output formats of the models, is also appreciated.

**Weaknesses:**

- In the released dataset, the choices are directly integrated into the prompt. It would be good to also add a column with only the original question, and another column containing the list with the possible options, so that researchers could evaluate their models with the prompts they used during their fine-tuning.
- As the authors mentioned, it would be useful to also create a test set for this benchmark.
- It would have probably made more sense to publish this in the Datasets and Benchmarks track.

**Questions:**

I personally noticed many hallucinations in the SEED benchmark, with some Q/A pairs that are simply false.
Since the Q/A pairs from SEED represent 28.3% of your dataset, I am worried that you would have such incorrect pairs in your benchmark.
Can you confirm that this was removed during the manual filtering?

**Limitations:**

Yes

---

> ### Author Rebuttal · Authors · 2024-08-06
>
> Thank you very much for your thorough review and appreciation of our work. Below, we address your concerns point by point:
>
> **Q1: In the released dataset, the choices are directly integrated into the prompt. It would be good to also add a column with only the original question, and another column containing the list with the possible options, so that researchers could evaluate their models with the prompts they used during their fine-tuning.**
>
> **A1:** Good point!  We will revise the format as per your suggestions for the public release of the benchmark. Here is a description of the revised columns for the benchmark:
> - **Index**: The sample number in the benchmark.
> - **Question**: Contains only the question itself without options.
> - **Image**: The content of the image.
> - **Options**: Lists the content of all options.
> - **Answer**: A single letter indicating the correct answer.
> - **Category**: Indicates the core dimension to which the sample belongs.
> - **L2 Category**: Indicates the detailed axis to which the sample belongs.
> - **Meta Info**: Indicates the source of the sample from previous multimodal benchmarks.
> We hope this format will facilitate evaluation within the LVLMs community.
>
> **Q2: As the authors mentioned, it would be useful to also create a test set for this benchmark.**
>
> **A2:** Thank you for your valuable feedback. We will strive to create a private test set in the future. We believe this test set can help the LVLMs community more comprehensively and fairly evaluate the actual multi-modal capabilities of existing models.
>
> **Q3: It would have probably made more sense to publish this in the Datasets and Benchmarks track.**
>
> **A3:** In fact, our work begins with two important and interesting observations regarding the evaluation of existing LVLMs. The first observation is that many samples in current benchmarks can be correctly answered without relying on visual content. The second observation is the phenomenon of data leakage hidden in the final evaluation scores of LVLMs. The second observation is derived from a detailed analysis of many carefully constructed experimental results, revealing the potential data leakage in LLMs and LVLMs that has not been adequately addressed in the current LVLM evaluation field.
>
> Based on these two observations, we propose two solutions. The first is a meticulously constructed benchmark that ensures all samples are visually dependent and are, as far as possible, not leaked in the training corpus of LLMs. However, this benchmark cannot entirely prevent some samples from being present in the training data of LVLMs. Additionally, it cannot guarantee that the samples will remain unexposed to LLMs and LVLMs introduced after the benchmark's creation date. Therefore, we also innovatively propose two supplementary metrics: Multi-modal Gain and Multi-modal Leakage.
>
> It is important to note that these two metrics are not exclusively tied to MMStar. They can be used to measure the extent of data leakage and the actual multimodal capability improvements gained from multimodal training in any benchmark. These metrics allow researchers to evaluate multimodal training leakage and benefit levels at any time, independent of the benchmark's creation date.
>
> Therefore, the contributions of this work include two important and intuitive observations, a benchmark, and a set of multimodal evaluation methodologies. After careful discussion, we believe this work is more suitable for the main track.
>
> **Q4: I personally noticed many hallucinations in the SEED benchmark, with some Q/A pairs that are simply false. Since the Q/A pairs from SEED represent 28.3% of your dataset, I am worried that you would have such incorrect pairs in your benchmark. Can you confirm that this was removed during the manual filtering?**
>
> **A4:** Astute observation! The occurrence of hallucinations in samples seems inevitable for large-scale benchmarks. Compared to the initial candidate pool of 14,000 samples from SEED, we retained only around 400 samples in our final benchmark, significantly reducing the cost of manual review. Therefore, all samples in MMStar underwent cross-validation by three experts to minimize the issue of hallucinations as much as possible.
>
> Your constructive comments and criticisms will greatly assist us in improving this work. Please do not hesitate to contact us if you have any further questions.

---

> > ### Comment · Reviewer_JQaJ · 2024-08-07
> > **Answer to authors**
> >
> > Thank you for answering the questions.

---

> > > ### Author Response · Authors · 2024-08-07
> > > **Thanks for the appreciation**
> > >
> > > Thank you for your prompt response and appreciation. Your suggestions have indeed helped us improve the quality of this work.

---

### Official Review · Reviewer_bz7V · 2024-07-13

**Soundness:** 3
**Presentation:** 3
**Contribution:** 2
**Rating:** 7
**Confidence:** 3

**Summary:**

In this paper, the authors examine current benchmarks for large vision-language models (LVLMs) and identify two main problems: 1) many samples do not require visual content, and 2) there is unintentional data leakage in LLM and LVLM training. To address these issues, they developed a multimodal benchmark called MMStar, consisting of 1,500 samples, and proposed two metrics to measure data leakage and performance gain in LVLMs’ multimodal training. They conducted empirical evaluations on 16 LVLMs to report their performance on MMStar.

**Strengths:**

1. The paper is well-organized and easy to follow.
2. The motivation behind the study is clear, and the empirical analysis is thorough.
3. Data curation for MMStar is comprehensively explained.
4. The proposed performance metrics are intuitive and effectively presented.

**Weaknesses:**

1. The authors only consider multiple-choice questions for the MMStar benchmark. Including a wider variety of well-curated questions without choices would be great.
2. Similar to Figure 2, the authors should provide the LLM Hit Rate for the MMStar benchmark.
3. What is the percentage distribution of the 1,500 samples across the four difficulty categories?

**Questions:**

Please respond to the points of weakness I mentioned above.

**Limitations:**

Yes

---

> ### Author Rebuttal · Authors · 2024-08-06
>
> We are encouraged to see that you found our work intuitive, containing extensive experiments and empirical analysis, and well-written. We have endeavored to address your concerns as follows:
>
> **Q1: The authors only consider multiple-choice questions for the MMStar benchmark. Including a wider variety of well-curated questions without choices would be great.**
>
> **A1:** Thank you for your suggestion. In the current version of the MMStar benchmark, we choose to use a multiple-choice question format for two reasons:
>
> 1. Most of the mainstream multi-modal benchmarks are in multiple-choice format. We carefully select samples from these benchmarks to construct a comprehensive, fully visually dependent benchmark, making MMStar a multiple-choice format as well.
> 2. The multiple-choice format allows for more objective evaluation, avoiding fluctuations caused by variations in LLM versions (differences in LLM capabilities). For instance, in FreeVA [1], it was observed that the results of some open-ended multimodal benchmarks were easily influenced by the version of the GPT API used.
>
> However, if the questions and prompts given to the language model are appropriate, open-ended questions without options can indeed better assess the capabilities of LVLMs. We are open to exploring this approach in future work.
>
> **Q2: Similar to Figure 2, the authors should provide the LLM Hit Rate for the MMStar benchmark.**
>
> **A2:** Thanks for your valuable feedback. The LLM Hit Rate of MMStar is 0%, which is significantly lower than the lowest LLM Hit Rate of 10.3% observed in the previous 6 benchmarks. This exceptionally low LLM Hit Rate is a result of our meticulously designed benchmark construction pipeline. In the first step, we only select samples from 6 existing benchmarks that are hit 2 times or fewer by 8 advanced LLMs. We have detailed the statistics of the hit counts in MMStar in the table below. As shown, all samples in MMStar have hit counts far less than 6, resulting in an LLM Hit Rate of 0%. We will add them to the supplementary materials of the next version.
>
> | Number of hits | Number of samples |
> |----------------|-------------------|
> | 0              | 848               |
> | 1              | 392               |
> | 2              | 260               |
>
> **Q3: What is the percentage distribution of the 1,500 samples across the four difficulty categories?**
>
> **A3:** Thank you for your thorough review and reminder. In the table below, we present the difficulty distribution of samples in MMStar. As shown, nearly 80% of the samples are answered correctly by at most half (8) of the LVLMs, with easy samples comprising less than 10% of the total. This highlights MMStar's focus on challenging samples that require advanced multimodal capabilities from LVLMs.
>
> | Difficulty level | Number of samples |
> |------------------|-------------------|
> | Tough (0-3)      | 532               |
> | Hard (4-7)       | 631               |
> | Moderate (8-11)  | 189               |
> | Easy (12-16)     | 148               |
>
> Your constructive comments and criticisms will greatly assist us in improving this work. Please do not hesitate to contact us if you have any further questions.
>
> [1] FreeVA: Offline MLLM as Training-Free Video Assistant

---

### Official Review · Reviewer_btYn · 2024-07-15

**Soundness:** 3
**Presentation:** 3
**Contribution:** 3
**Rating:** 6
**Confidence:** 4

**Summary:**

The authors have identified two primary concerns with the benchmarks commonly used for large vision-language models (LVLMs). Firstly, many samples do not require visual content to answer the questions. Secondly, they noted unintentional data leakage during LVLM training. They assessed eight large language models (LLMs) across six widely-used multi-modal LLM benchmarks, demonstrating that LLMs can correctly answer a significant portion of questions without visual input. To more reliably evaluate LVLM performance, they developed a new benchmark by meticulously filtering data from six existing benchmarks with three requirements: 1) visual dependency, 2) minimal data leakage and 3) multi-modal capability for resolutions. Additionally, they designed two metrics: Multi-modal Gain, to quantify the improvement from multi-modal training, and Multi-modal Leakage, to assess the extent of potential data leakage. Using this new benchmark and the two metrics, they provide a comprehensive comparison of state-of-the-art LVLMs.

**Strengths:**

- The paper is well-structured and clearly articulated, facilitating ease of comprehension.

- The evaluation process is meticulously designed, and the conclusions drawn from it are convincing.

- The findings presented in this paper meaningfully impact multi-modal large language model (LLM) research. Researchers have depended heavily on benchmarks without thoroughly examining their quality. The authors question the reliability of evaluations based on these benchmarks. Without a reliable benchmark, it is impossible to faithfully measure actual multi-modal gain. They developed a new benchmark, MMStar, which facilitates more reliable evaluations.

- Using the MMStar benchmark, the authors evaluated two closed-source and fourteen open-source large vision-language models (LVLMs), with the results presented in Table 3. As expected, GPT4 emerged as the top performer in five out of six tasks. Additionally, they underscored the efficacy of smaller-scale models by highlighting that TinyLLaVa, a model with 3 billion parameters, outperformed some larger competitors with 7 billion and 13 billion parameters, thereby emphasizing the potential of smaller-scale LVLMs.

**Weaknesses:**

- The proposed metrics, Multi-modal Gain and Multi-modal Leakage, are dependent on the base LLM utilized in the large vision-language models. This dependency complicates the use of these metrics for directly comparing the multi-modal gain across different LVLMs.

- The manual review step aggressively reduces the MMStar benchmark from 11,607 samples to 1,500 samples. The explanation provided in Section 3.1 for this reduction is somewhat vague and lacks clear, objective criteria for filtering. I am curious about the rationale behind such an aggressive reduction by nearly tenfold. Is this reduction due to a scarcity of data meeting the three specified criteria mentioned between line 187 to 189, or are there other reasons for this decision?

**Questions:**

- Please clarify the goal of using proposed metrics

- Please clarify the decision to aggressively reduce the size of final benchmark

**Limitations:**

The authors properly discuss the limitation in the paper

---

> ### Author Rebuttal · Authors · 2024-08-06
>
> We thank you for the positive comments on the novelty and meaningful impact of our findings and proposed benchmark. We detail your concerns and our corresponding responses below:
>
> **Q1: The proposed metrics, Multi-modal Gain (MG) and Multi-modal Leakage (ML),  are dependent on the base LLM utilized in the large vision-language models. This dependency complicates the use of these metrics for directly comparing the multi-modal gain across different LVLMs.**
>
> **A1:** In fact, when developers want to evaluate the actual multi-modal capabilities of their LVLMs, they can directly choose our MMStar benchmark and perform inference with LVLM for quick evaluation. Furthermore, ML and MG can reflect the extent of data leakage during multimodal training and the actual improvement in the model's multimodal capabilities. We have already provided the performance of several popular LLM bases in our work, and we will continue to update with new LLM bases such as LLaMA3 and Gemma2 on MMStar. This will facilitate the community in quickly evaluating and comparing the multimodal gains of their models.
>
> Additionally, the proposed MG and ML metrics can serve as probes for the training corpus of LVLMs. For instance, averaged over seven benchmarks, ShareGPT4V-7B, compared to LLaVA-1.5-7B, significantly increases its MG by incorporating high-quality image-caption data under the same architecture without affecting ML, demonstrating the importance of high-quality image-caption data. Similarly, with an average of over 16 models, MMMU exhibited the lowest average MG, indicating minimal overlap between the multimodal training data of LVLMs and the samples in MMMU.
>
> **Q2: The manual review step aggressively reduces the MMStar benchmark from 11,607 samples to 1,500 samples. The explanation provided in Section 3.1 for this reduction is somewhat vague and lacks clear, objective criteria for filtering. I am curious about the rationale behind such an aggressive reduction by nearly tenfold. Is this reduction due to a scarcity of data meeting the three specified criteria mentioned between lines 187 to 189, or are there other reasons for this decision?**
>
> **A2:** Thank you for pointing out the lack of clarity in our description of the manual review stage. We provide a detailed supplement on this stage here and will integrate these details into the main text. After roughly filtering the original data pool with 8 advanced LLMs, resulting in 11,607 candidate samples, we initiate a rigorous manual review phase.
>
> First, we establish 6 core evaluation dimensions and 18 detailed axes by integrating the evaluation dimensions from existing benchmarks. Next, we use 16 LVLMs to infer and count the number of hits for each sample. Furthermore, we design a UI interface listing the current sample's image, options, answer, sample source, hit count, and the 18 detailed axes. The samples are arranged in ascending order based on the number of hits.
>
> The formal manual selection and benchmark construction process is as follows:
>
> 1. Preliminary Classification: Three experts are each responsible for two core capability dimensions (i.e., 6 detailed axes). They need to review all candidate samples and select and correctly classify the samples belonging to their respective dimensions. The samples selected must retain their visual dependency.
> 2. Statistical Analysis: After the preliminary classification, we consider the numerical balance between dimensions and the difficulty level of the samples. Samples under the "coarse perception" dimension approach 4,000, while those under "logical reasoning" are fewer than 700. In terms of difficulty distribution, there are 4,555 easy (i.e., number of hits between 12 and 16) samples but only 2,758 tough (i.e., number of hits between 0 and 3) ones. Given these premises, a lot of  repetitive simple samples, such as those merely asking for the color of an object in the image, are not what we desire.
> 3. Initial Benchmark: After considering both the numerical balance and difficulty level of the samples, we set the total sample number of the benchmark at 1,500, with each core capability dimension containing 250 samples. Then, we assign each expert two core capability dimensions, instructing them to prioritize sample difficulty when selecting 250 samples per dimension.
> 4. Cross-Validation: To minimize personal bias, we arrange for each expert to review the dimensions handled by the other two experts after the initial benchmark is constructed. Samples with issues are replaced by correct samples of the same difficulty level from the candidate pool.
> By following this thorough process, we ensure a balanced and challenging benchmark set.
>
> If you still have any concerns or aspects you would like to discuss further, please do not hesitate to contact us at any time.

---

### Author Rebuttal · Authors · 2024-08-06

We sincerely appreciate all reviewers for your time and efforts in the review. All detailed questions of each reviewer are answered accordingly in each column below. We hope these responses can address the reviewers' concerns adequately. Additionally, we provide the implementation details of the manual review process used in constructing MMStar to supplement the missing details in the manuscript.

After roughly filtering the original data pool with 8 advanced LLMs, resulting in 11,607 candidate samples, we initiate a rigorous manual review phase. First, we establish 6 core evaluation dimensions and 18 detailed axes by integrating the evaluation dimensions from existing benchmarks. Next, we use 16 LVLMs to infer and count the number of hits for each sample. Furthermore, we design a UI interface listing the current sample's image, options, answer, sample source, hit count, and the 18 detailed axes. The samples are arranged in ascending order based on the number of hits.

The formal manual selection and benchmark construction process is as follows:

1. Preliminary Classification: Three experts are each responsible for two core capability dimensions (i.e., 6 detailed axes). They need to review all candidate samples and select and correctly classify the samples belonging to their respective dimensions. The samples selected must retain their visual dependency.
2. Statistical Analysis: After the preliminary classification, we consider the numerical balance between dimensions and the difficulty level of the samples. Samples under the "coarse perception" dimension approach 4,000, while those under "logical reasoning" are fewer than 700. In terms of difficulty distribution, there are 4,555 easy (i.e., number of hits between 12 and 16) samples but only 2,758 tough (i.e., number of hits between 0 and 3) ones. Given these premises, a lot of  repetitive simple samples, such as those merely asking for the color of an object in the image, are not what we desire.
3. Initial Benchmark: After considering both the numerical balance and difficulty level of the samples, we set the total sample number of the benchmark at 1,500, with each core capability dimension containing 250 samples. Then, we assign each expert two core capability dimensions, instructing them to prioritize sample difficulty when selecting 250 samples per dimension.
4. Cross-Validation: To minimize personal bias, we arrange for each expert to review the dimensions handled by the other two experts after the initial benchmark is constructed. Samples with issues are replaced by correct samples of the same difficulty level from the candidate pool.

Moreover, we also provide the number of samples with consensus before and after the cross-validation step in the manual review process for MMStar in the table below. Only samples that all three experts unanimously agree upon are retained; otherwise, they are replaced with samples of the same difficulty level from the candidate pool.

|          | Before | After |
|----------|--------|-------|
| Expert 1 | 472    | 500   |
| Expert 2 | 468    | 500   |
| Expert 3 | 483    | 500   |

By following this thorough process, we ensure a balanced and challenging benchmark set.

---

### Decision · Program_Chairs · 2024-09-25

**Decision:**

Accept (poster)

**Comment:**

This submission focuses on large vision-language models.  The authors have identified several problems posed by conventional datasets, and have developed a new multimodal benchmark and proposed two metrics providing a new framework for better evaluating these multimodal models.  All the reviewers emphasized the originality of the proposed work.
They also asked for several clarifications.  After the rebuttal and discussion session, all were satisfied with the answers.  All reviewers were positive about the submission, considering that the novelty and the interest of the proposal stand out clearly.
The authors are strongly encouraged to take all comments into account in their final version.